# Real-time monitoring of hydrophobic aggregation reveals a critical role of cooperativity in hydrophobic effect

Liguo Jiang[1,2,3,*], Siqin Cao[2,3,*], Peter Pak-Hang Cheung[4], Xiaoyan Zheng[2], Chris Wai Tung Leung[2], Qian Peng[5], Zhigang Shuai[6], Ben Zhong Tang[2,3,4,7], Shuhuai Yao[3,7,8] & Xuhui Huang[2,3,7]

The hydrophobic interaction drives nonpolar solutes to aggregate in aqueous solution, and hence plays a critical role in many fundamental processes in nature. An important property intrinsic to hydrophobic interaction is its cooperative nature, which is originated from the collective motions of water hydrogen bond networks surrounding hydrophobic solutes. This property is widely believed to enhance the formation of hydrophobic core in proteins. However, cooperativity in hydrophobic interactions has not been successfully characterized by experiments. Here, we quantify cooperativity in hydrophobic interactions by real-time monitoring the aggregation of hydrophobic solute (hexaphenylsilole, HPS) in a microfluidic mixer. We show that association of a HPS molecule to its aggregate in water occurs at sub-microsecond, and the free energy change is $-5.8$ to $-13.6\,\mathrm{kcal\,mol^{-1}}$. Most strikingly, we discover that cooperativity constitutes up to 40% of this free energy. Our results provide quantitative evidence for the critical role of cooperativity in hydrophobic interactions.

[1] Institute for Advanced Study, The Hong Kong University of Science and Technology, Hong Kong, China. [2] Department of Chemistry, The Hong Kong University of Science and Technology, Hong Kong, China. [3] HKUST-Shenzhen Research Institute, Hi-Tech Park, Nanshan, Shenzhen 518057, China. [4] Division of Biomedical Engineering, The Hong Kong University of Science and Technology, Hong Kong, China. [5] Key Laboratory of Organic Solids, Institute of Chemistry, Chinese Academy of Sciences, Beijing 100190, China. [6] Department of Chemistry, Tsinghua University, Beijing 100084, China. [7] Hong Kong Branch of Chinese National Engineering Research Center for Tissue Restoration & Reconstruction, The Hong Kong University of Science and Technology, Hong Kong, China. [8] Department of Mechanical and Aerospace Engineering, The Hong Kong University of Science and Technology, Hong Kong, China. * These authors contributed equally to this work. Correspondence and requests for materials should be addressed to S.Y. (email: meshyao@ust.hk) or to X.H. (email: xuhuihuang@ust.hk).

Hydrophobic interactions drive nonpolar solutes to aggregate in aqueous solution[1,2], and hence play an important role in many fundamental processes in nature. Molecular theories[3–8] and computer simulations[9,10] of hydrophobicity suggest that water molecules need to break some of their hydrogen bonds to accommodate large nonpolar solutes (radius larger than 1 nm). This loss of water–water hydrogen bonds will induce fluctuations and depletion of water density near large hydrophobic solutes[10–13], and further lead them to collapse[14,15]. Therefore, hydrophobic interactions are originated from the collective motions of water hydrogen bond networks surrounding hydrophobic solutes[16–19]; hence, they are naturally cooperative[20–22]. This is in contrast to fundamental intermolecular interactions that are often treated as pairwise additive, such as ionic interactions, dipolar interactions and dispersion forces.

Cooperativity is a phenomenon whereby the overall interaction for a system containing multiple molecules is stronger than the summation of individual pairwise interactions. Such cooperative feature of hydrophobic interactions is widely believed to accelerate, stabilize and enhance the formation of hydrophobic core in proteins, the aggregation of misfolded proteins and the formation of lipid vesicles and micelles. However, the extent of contributions from cooperativity to these processes still remains unclear. Addressing this issue requires, first and foremost, quantitative measurements of hydrophobic interactions, which are very challenging as the size, chemistry and topography of solutes can affect the strength of hydrophobic interactions[3,23–27].

Microfluidic mixing techniques have been widely applied to investigate many important chemical and biological processes, including protein and RNA folding[28–32], enzyme activities[33,34] and vesicle formations[35,36]. In this study, we quantified the thermodynamics and kinetics of hydrophobic interactions in bulk solution by real-time monitoring of fluorescence induced by the aggregation of hexaphenylsilole[37] (HPS, $C_{40}H_{30}Si$) in a state-of-the-art microfluidic mixer at microsecond timescale. Using this technique, we show that in the attachment of a HPS molecule to its aggregate, the free energy change is $-5.8$ to $-13.6$ kcal mol$^{-1}$, the timescale is sub-microsecond, and the cooperativity constitutes up to 40% of the free energy change.

## Results

**Microfluidic experiment and model fitting.** We employed HPS molecules (Fig. 1a left) to study hydrophobic interactions in bulk solutions using a microfluidic mixer as depicted in Fig. 1b (Supplementary Note 1). The aggregation of HPS molecules is mainly driven by solvent-induced hydrophobic interactions, as six aromatic rings render its hydrophobic property and direct $\pi - \pi$ stacking interactions between HPS molecules are negligible[38]. Importantly, HPS molecules emit strong fluorescence upon aggregation[37]. This unique feature allows us to track the progress of HPS aggregation in the microfluidic mixer (Fig. 2a), where the molecular aggregation occurs in a sample stream that was hydrodynamically sheathed to tens of nanometres in width within a few microseconds upon rapid solvent exchange. We can determine the time course of HPS aggregation via dividing the travelling distance of mixture solution along the exit microchannel by its flow velocity. The measured fluorescence intensities (Fig. 2b,c, symbol points) were linearly correlated with the total amount of aggregated HPS. This linear correlation was validated by quantum mechanics/molecular mechanics calculations[39], fluorescence AFM experiments and spectrophotometer experiments (Supplementary Figs 1–4, 13, 19 and 20; Supplementary Notes 2 and 6).

Next, we fitted the measured fluorescence data to the classical nucleation-growth model[40] (Fig. 3a; Supplementary Note 3). At each aggregation time point in this model, new nuclei are being formed and simultaneously existing aggregates are growing. Once new nuclei are formed, they continue to grow as long as the solution is supersaturated. Accordingly, the total amount of aggregated HPS per unit volume at time $t$ is the integration of the product of nuclei generated at a previous time $s$ and its size growing during the remaining time $\tau$

$$V(t) = \int_0^t J(s) \left( n^\star(s) + \int_s^t \frac{dg(\tau', s)}{d\tau'} d\tau \right) ds \qquad (1)$$

where $J(s)$ is the number of nucleus generated in unit volume per time, $n^\star(s)$ is the critical nucleus size, and $dg(\tau', s)/d\tau'$ is the growth rate of the nucleus that formed at time $s$. By fitting the measured fluorescence to the model (Fig. 2b,c, solid lines), we obtained the free energy change associated with attaching a HPS monomer to its aggregate and resolved the kinetics of HPS aggregation in various solvent mixtures of dimethyl sulfoxide (DMSO) and water. The results were then extrapolated to the pure water condition (Supplementary Fig. 5).

**Free energy change associated with hydrophobic aggregation.** We determined that the strength of hydrophobic interaction associated with attaching a HPS monomer to an existing aggregate (for example, 5.8–13.6 kcal mol$^{-1}$ in pure water, Fig. 1c) is comparable to that of several water–water hydrogen bonds[41]. As shown in Figs 1c and 3b, hydrophobic interaction increases rapidly with aggregate size and decreases with the addition of DMSO solvent. For instance, the free energy of hydrophobic interaction by attaching a HPS monomer to an aggregate containing 10 molecules ($\sim 29$ Å diameter) is $-8.2$ kcal mol$^{-1}$ in pure water, while it is only $-3.8$ kcal mol$^{-1}$ in the solvent mixture with a DMSO mole fraction of 0.32. As the aggregate grows in size, the hydrophobic interactions become stronger; when the aggregate is infinitely large, they reach the asymptotic values of $-13.6$ and $-6.7$ kcal mol$^{-1}$ in pure water and in the solvent mixture, respectively. In addition, the hydrophobic free energy per solvent accessible surface area (SASA) of a HPS molecule (SASA: 750 Å$^2$) in pure water was calculated to be 18.1 cal mol$^{-1}$ Å$^{-2}$, a value close to the estimated one ($\sim 20$ cal mol$^{-1}$ Å$^{-2}$) for aromatic hydrocarbons from previous experiment[42] and the predicted one ($\sim 16$ cal mol$^{-1}$ Å$^{-2}$) for benzene from MD simulations[43]. Interestingly, Chandler and co-workers predicted that there should exist a crossover in the length-scale for the hydrophobic effect at around 1 nm in solute radius[1,3]. Our experiment demonstrates the existence of this crossover by showing a kink during the transition from volume-based hydration free energy for monomer (smaller than 1 nm in radius) to area-based hydration free energy of aggregates (larger than 1 nm in radius, see Supplementary Note 7 and Supplementary Fig. 21 for details).

**Cooperativity of hydrophobic interactions.** We found that cooperativity constitutes up to 40% of the free energy for the hydrophobic association in aggregate formation. To compute the magnitude of cooperativity, we followed Wang *et al.*[21] to define it as the excess multibody free energy upon attaching a monomer to the aggregate (Fig. 3b). Figures 1c and 3b show the dependence of cooperativity on the size of hydrophobic aggregate in different solvent conditions. The strength of cooperativity in attaching a HPS monomer to aggregates increases from 2.6 kcal mol$^{-1}$ for aggregate containing five molecules ($\sim 23$ Å diameter) to 4.2 kcal mol$^{-1}$ for aggregate containing 50 molecules ($\sim 45$ Å diameter) in pure water, whereas the free energy change

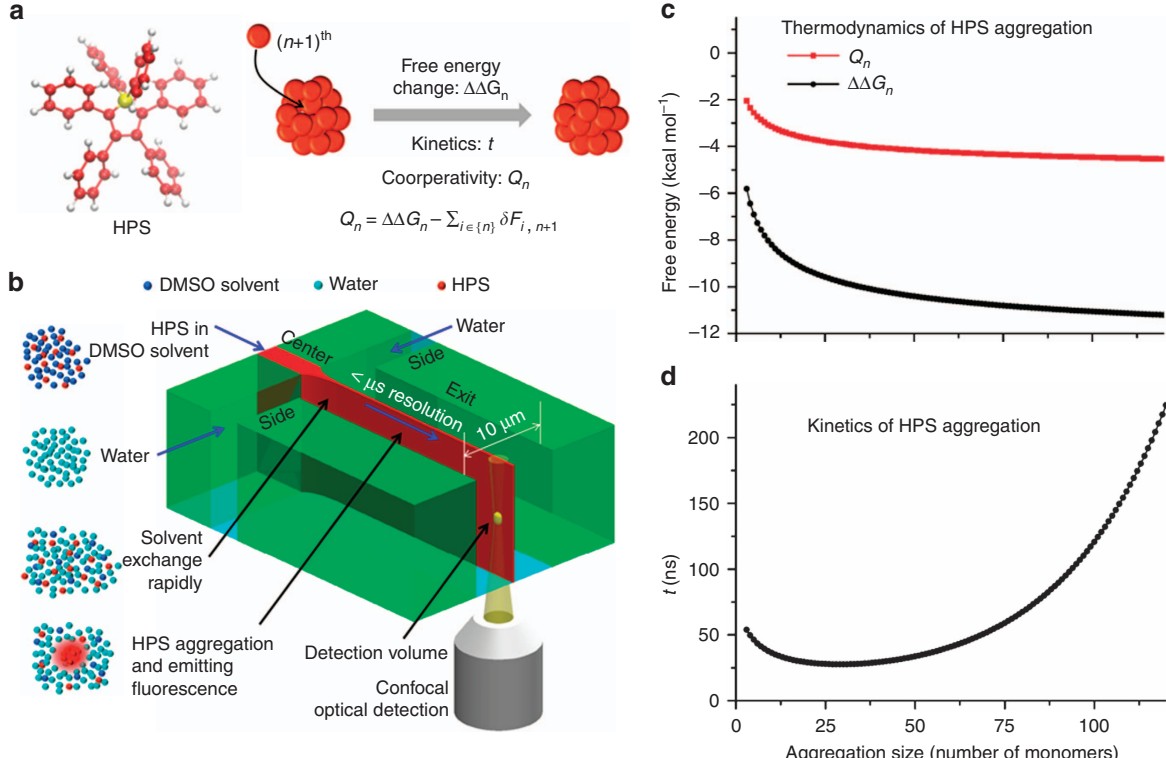

**Figure 1 | Quantifying cooperativity in hydrophobic interactions by monitoring HPS aggregation.** (**a**, left) The chemical structure of a HPS molecule. (right) A scheme of HPS aggregation thermodynamics and kinetics. Cooperativity ($Q_n$) is defined as the difference between the associated free energy of attaching a HPS monomer to the aggregate ($\Delta\Delta G_n$) and the summarization of the two-body potential of mean force $\left(\sum_{1\le i\le n}\delta F_{i,n+1}\right)$ upon the attachment of the $(n+1)^{th}$ monomer to the aggregate. (**b**) The principle of real-time monitoring of HPS aggregation in the microfluidic mixer. HPS (red dots) dissolved in DMSO (blue dots) is continuously pumped into the centre microchannel, and then squeezed by two side water (cyan dots) streams to form an extremely narrow stream with tenths of nanometres in width. Thus, rapid solvents exchange occurs in a pure diffusion manner, and HPS molecules aggregate in downstream with strong fluorescence emission under confocal optical microscopy. Blue arrows indicate the directions of continuous fluid flow. Black arrows indicate specific locations. (**c**,**d**) Thermodynamics ($\Delta\Delta G$) (**c**), cooperativity ($Q_n$) (**c**) and kinetics ($t$) (**d**) of attaching a HPS monomer to aggregates in pure water.

associated with these processes are –6.9 and –10.4 kcal mol$^{-1}$, respectively. To evaluate the total amount of cooperativity in the formation of HPS aggregate in pure water, we integrated the contribution of cooperativity in Fig. 1c and compared this accumulated cooperativity with the aggregate formation free energy (that is the integrated hydrophobic interactions in Fig. 1c). As shown in Supplementary Fig. 6, both the aggregate formation free energy and the accumulated cooperativity increase linearly with aggregate size. Interestingly, our reported hydrophobic free energy at large $n$ limit $\left(\Delta\Delta G_{n\to\infty}=-13.6\,\text{kcal mol}^{-1}\right)$ becomes equivalent to the free energy of transferring a HPS molecule from water to the HPS phase. Importantly, cooperativity constitutes up to 40% of the formation free energy in the aggregation process. This finding demonstrates a critical role of cooperativity in hydrophobic aggregation.

**Kinetics of hydrophobic aggregation.** Through our kinetics analysis, the attachment of a HPS molecule to an aggregate was estimated to occur at tens to hundreds of nanoseconds. We investigated the kinetics of HPS aggregation by tracking the growth of the first nucleus formed in the theoretical model (Supplementary Fig. 7). At the initial stage of aggregation when aggregate size is smaller than 10 molecules, the time required for attaching a monomer to aggregates ranges from 30 to 200 ns (Figs 1d and 3c). When the aggregate grows in size, the kinetics of hydrophobic aggregation first accelerate with increased surface area available for attaching, and then gradually slow down due to the depletion of HPS monomers. Also, the time required for HPS molecules in pure water to form an aggregate containing 50 molecules ($\sim$45 Å diameter) is 1.9 µs (Supplementary Fig. 7). In addition, the initial HPS concentration also influences the aggregation kinetics (Supplementary Fig. 8). For instance, at an initial HPS concentration of 2 mM in the solvent with 0.16 DMSO mole fraction, the time required for monomers to form an aggregate containing 50 molecules is 11.8 µs (Supplementary Fig. 7). Our observation of microsecond timescale of HPS aggregation is comparable to the timescale of early stage protein folding (see the Methods section for details), during which nonspecific hydrophobic interactions among hydrophobic residues induce the initial collapse of polypeptide chain at microsecond timescale[44].

**Discussions**

Organic molecules, such as proteins and lipid, bury their hydrophobic components to form stable cores. Hydrophobic interaction plays a crucial role in facilitating the collapse of protein chains into a globular shape[25,45–47]. The faster kinetics of hydrophobic aggregations (at microsecond), in contrast to protein folding (at millisecond or longer), suggest that the formation of protein cores by the aggregation of hydrophobic side-chains occurs at the early stage in the process of globular protein folding. Most importantly, we show that hydrophobic

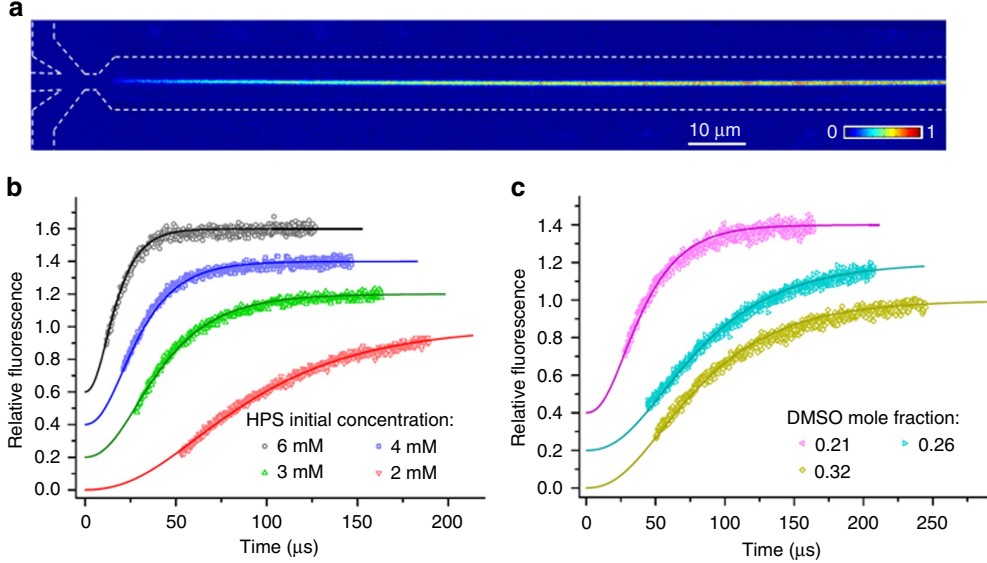

**Figure 2 | Time evolution of fluorescence intensity measured by experiment and fitted by theory.** (**a**) A representative fluorescence image of HPS aggregation in the microfluidic mixer (with subtraction of background fluorescence). White dashed lines indicate the outline of microfluidic mixer. (**b**) Kinetic profiles of HPS aggregation at various initial HPS concentrations (the solvent condition: DMSO mole fraction of 0.16). The solute–solvent surface tension ($\gamma_{sl}$) through theoretical fitting (solid lines) of experimental data (symbol points) are 20.9 ($\pm$0.6), 20.7 ($\pm$0.6), 20.5 ($\pm$0.6), and 20.3 ($\pm$0.6) cal mol$^{-1}$Å$^{-2}$ at initial HPS concentration of 6, 4, 3 and 2 mM, respectively. (**c**) Kinetic profiles of HPS aggregation in various solvent conditions (initial HPS concentration of 6 mM). For clear illustrations, the relative fluorescence curves in part (**b**) corresponding to HPS concentrations of 3, 4 and 6 mM are shifted along y axis by 0.2, 0.4 and 0.6, respectively. Similarly, relative fluorescence curves in part (**c**) corresponding to DMSO mole fractions of 0.26 and 0.21 are shifted by 0.2 and 0.4, respectively. The solute–solvent surface tension ($\gamma_{sl}$) through theoretical fitting (solid lines) of experimental data (symbol points) are 19.5 ($\pm$0.6), 18.5 ($\pm$0.6) and 17.0 ($\pm$0.6) cal mol$^{-1}$Å$^{-2}$ in the solvent condition with a DMSO mole fraction of 0.21, 0.26 and 0.32, respectively. The Pearson correlation coefficients for all fitted curves are larger than 0.98 (Supplementary Fig. 11).

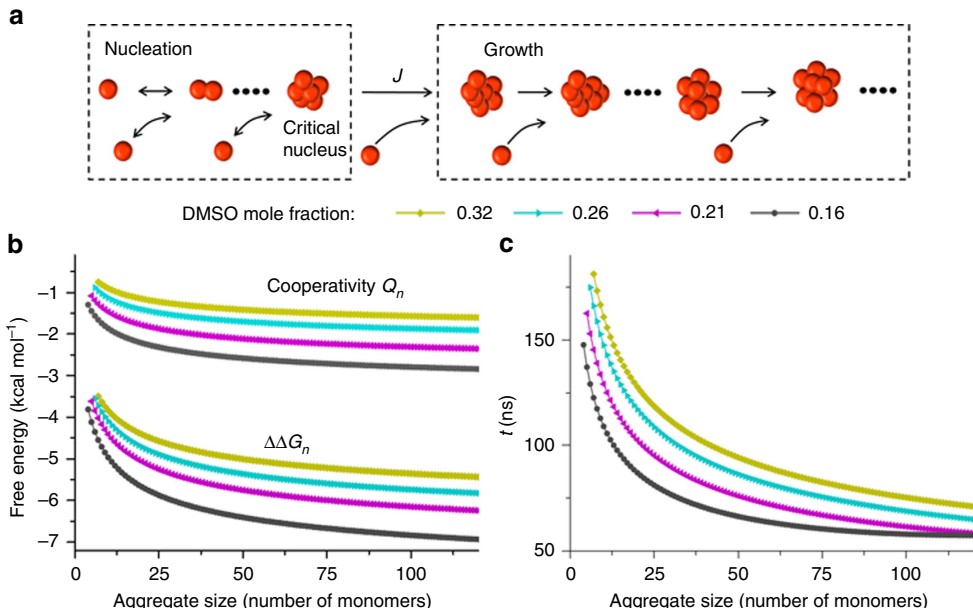

**Figure 3 | Thermodynamics and kinetics as well as cooperativity of HPS aggregation.** (**a**) A schematic illustration of the nucleation-growth model for HPS aggregation. At each aggregation time point in this model, new nuclei are being formed at a rate of $J$ determined by the classical nucleation theory, and then are growing in size in a barrier-free and diffusion-controlled manner. (**b**) The free energy change and cooperativity associated with attaching a HPS monomer to aggregates in various DMSO/water solvent mixtures. (**c**) The time taken for aggregates to grow by one HPS molecule in various solvent mixtures at initial HPS concentration of 6 mM.

interaction, which is an interaction induced by collective behaviours of many water molecules, is strongly cooperative, and thus substantially enhance its strength during the aggregation

of dispersed hydrophobic molecules in solution. We anticipate that our findings have profound implications in protein folding, as the protein core formation involves the collapse of

hydrophobic side-chains. We acknowledge that these two processes are different in many aspects. For instance, flexible protein chains contain numerous conformations, while HPS molecules are relatively rigid. In addition, multiple sources may contribute to cooperativity in protein folding, such as the cooperative helix melting process due to hydrogen bonding[48]. Furthermore, even the extent of contributions by hydrophobic interactions to protein folding remains elusive[49]. In spite of these differences, our findings highlight the important role of hydrophobic cooperativity (as large as 40%) in the initial collapse of protein chain to form into a globular shape. We expect that our experimental platform will have promising applications in studying hydrophobic interactions of a wide range of organic molecules with aggregation-induced emission[50], and in investigating the impact of important factors such as temperature on hydrophobic effect.

## Methods

**Microfluidic experiments.** A solution of HPS molecules dissolved in DMSO was continuously pumped into the centre microchannel. This central stream was hydrodynamically squeezed by two side water streams to form an extremely narrow stream with tenths of nanometres in width. Hence, rapid solvents exchange occurred in a pure diffusion manner, and the immediate environment for HPS aggregation was reached within a few microseconds. In the mixer, the time course of HPS aggregation was determined by dividing the travelling distance of mixture solution along the exit microchannel by its flow velocity. Thus, the progress of HPS aggregation in downstream was monitored by the integrated confocal system at sub-microsecond temporal resolution. Fluorescence images were captured with spatial resolutions of 1 and 0.5 μm in vertical (depth) and horizontal (width) directions, respectively. A diode laser at 405 nm was employed as the excitation source. The excitation beam was focused into the centre layer of microchannels by an oil immersion objective lens ( × 63/1.4 NA). Then, fluorescence was collected by the same objective and detected by a photomultiplier tube detector at wavelength of 420–600 nm. Pleases refer to Supplementary Note 1 for more details of the microfluidic mixing experimental setup.

**The classical nucleation-growth theory.** In this paper, we employed the classical nucleation-growth theory[40] to model the time evolution of the total amount of aggregated HPS in solution. At each aggregation time point in this model, new nuclei are being formed and the rate of nuclei formation is determined by the classical nucleation theory (CNT). At the same time, existing aggregates are growing in size in a barrier-free and diffusion-controlled manner. Once new nuclei are formed, they continue to grow as long as the solution is supersaturated.

The nucleation work $W(n)$ to form an aggregate containing $n$ molecules, the critical nucleus size $n^*$ and the nucleation rate $J(t)$, which is the number rate of new nuclei generated per volume per time, are obtained from the CNT[40] with the assumption that HPS monomer and HPS aggregates are spherical as suggested by previous studies[51]:

$$W(n)=\gamma_{sl}A_n - nkT\ln S(t)=\gamma_{sl}(36\pi)^{\frac{1}{3}}\left(\frac{v}{p}\right)^{\frac{2}{3}}n^{\frac{2}{3}} - nkT\ln S(t) \quad (2)$$

$$n^*=\left(\frac{2\gamma_{sl}(36\pi)^{1/3}(v/p)^{2/3}}{3KT\ln S(t)}\right)^3=\left(\frac{2\theta}{3\ln S(t)}\right)^3 \quad (3)$$

$$J(t)=\left[(36\pi)^{1/6}\frac{DC_e}{\sqrt[3]{v^2/p^2}\sqrt{\theta}}\right]S(t)\ln S(t)e^{-4\theta^3/(27\ln^2 S(t))} \quad (4)$$

where $\gamma_{sl}$ is the surface tension of HPS aggregates in the solvent mixture, $A_n=(36\pi)^{1/3}(nv/p)^{2/3}$ is the surface area of an aggregate containing $n$ molecules, $k$ is the Boltzmann constant, $T$ is the temperature, $S(t)=C_1/C_e$ is the supersaturation ratio of HPS monomer in the solvent mixture changing with time $t$, $v$ is the molecular volume of HPS monomer, $p$ is the packing density of aggregates, which is the fraction of the molecular volume over the average volume that is occupied by a molecule in the aggregate, $D$ is the diffusion coefficient of HPS molecule, $C_e$ is the solubility of HPS in the solvent mixture and $\theta=\gamma_{sl}(36\pi)^{1/3}(v/p)^{2/3}/kT$ is related to the surface energy of aggregates. We adopted the same solute–solvent surface tension $\gamma_{sl}$ for aggregates in the same solvent mixture.

The HPS molecular volume, the packing density and the diffusion constant of HPS molecule were obtained as follows: by assuming that HPS molecules are spherical, the volume of HPS molecule can be calculated as $v=4\pi R_1^3/3$, where $R_1$ is the molecular radius. For spherical aggregate, the packing density was chosen to be $p = 0.7$ as suggested by previous studies[43]. The diffusion constant of HPS monomer $D$ was calculated from the Einstein–Stokes relation, $D=kT/(6\pi\eta R_1^D)$, where $\eta$ is the viscosity of solvent mixtures obtained from the reported data[52], and $R_1^D$ is the

diffusive radius of HPS monomer. The molecular radius $R_1$ and the diffusive radius $R_1^D$ of HPS molecule were computed from the surface area of the HPS molecule with $4\pi R_1^2=A_{HPS}$ and $4\pi(R_1^D + R_W)^2=A_{HPS}^{SASA}$. Here $R_W = 1.4$ Å is the radius of water molecule, $A_{HPS}$ is the surface area of HPS molecule, and $A_{HPS}^{SASA}$ is the SASA of HPS molecule. With DFT calculations using the B3LYP functional[53–55] and 6–31 g basis set[56], we optimized the structure of HPS molecule (Fig. 1a, left), and obtained $A_{HPS} = 437$ Å$^2$ and $A_{HPS}^{SASA} = 750$ Å$^2$. These values give the radius of the HPS molecule $R_1 = 5.9$ Å and the diffusive radius $R_1^D = 6.3$ Å, respectively. The calculated diffusion constants of HPS monomer in different solvent mixtures are listed in Supplementary Table 1.

For the growth of HPS aggregates, we adopted a diffusion-controlled kinetic model in the absence of activation free energy barrier for HPS molecules attaching to an existing aggregate, and this model has been adopted by previous studies[57]. In our model, the monomer attachment frequency $f_n$ is the product of two terms: the flux of incoming molecules to encounter the surface of the existing aggregate containing $n$ molecules, and the surface area of the aggregate:

$$f_n(t)=\frac{DC_1(t)}{R_n} \cdot 4\pi R_n^2=4\pi DC_1(t)\cdot\left[\frac{3n(t)v}{4\pi p}\right]^{1/3} \quad (5)$$

where $D$ is the monomer diffusion constant, $C_1(t)$ is the monomer concentration at time $t$, $R_n$ is the radius of the aggregate containing $n$ molecules and $n(t)$ is the size of the aggregate at time $t$. The molecular detachment frequency is independent of monomer concentration[58], and is approximated to the attachment frequency at equilibrium $f_n^e$. Hence, the growth rate of an aggregate containing $n$ molecules can be written as

$$\frac{dg(t)}{dt}=f_n(t) - f_n^e(t)=4\pi DC_e\left[\frac{n(t)v/p}{4\pi/3}\right]^{1/3}[S(t) - 1] \quad (6)$$

**Fitting fluorescence data to the classical nucleation-growth model.** We fitted the experimental measured fluorescence data to a theoretical model, which was built upon the classical nucleation-growth theory (equations (2)–(5)). In our model, new nuclei are being formed at each aggregation time point, and simultaneously existing aggregates are growing. Once new nuclei are formed, they continue to grow as long as the solution is supersaturated. Accordingly, the total amount of aggregated HPS ($V(t)$) per unit volume at time $t$ is the integration of the product of nuclei generated at a previous time $s$ and its size growing during the remaining time $\tau$. Meanwhile, the total amount of aggregated HPS ($V(t)$) can be independently calculated from supersaturation ratio ($S(t)$, see Supplementary Note 3; Supplementary equation (8) for details). On the basis of these relations at a given initial concentration ($C_0$), the solution of $V(t)$ can be obtained numerically. As we showed that the total volume of aggregated HPS is proportional to the total fluorescence intensity, the measured dynamics of normalized fluorescence $I(t)/I_T$ should directly correlate to the HPS aggregation kinetics $V(t)/V_T$. Therefore, we used the normalized total aggregates volume $V(t)/V_T$ to directly fit the measured normalized fluorescence intensity. In our model, $\gamma_{sl}$, the surface tension of HPS aggregates in the solvent mixture, is the only fitting parameter. As our quantitative analysis heavily relies on the CNT, we have examined an alternative theory: a non-CNT[59,60] involving stable prenucleation clusters (Supplementary Note 4). In non-CNT, the existence of stable prenucleation clusters will introduce a second free energy barrier, ε (Supplementary Note 4), in comparison with CNT containing only a single barrier. Applying non-CNT to our system, we show that ε has to be within $1.2kT$ (comparable with thermal fluctuations) in order to obtain reasonable fitting to experimental fluorescence (Supplementary Figs 14 and 15). Not surprisingly, with these small values of ε, the non-CNT theory provides consistent results with our CNT theory in both $\Delta\Delta G$ and cooperativity (Supplementary Fig. 16). These results suggest that the formation of stable prenulceation clusters is not substantial in the aggregation of HPS molecules, and thus CNT is well applicable to our system. In our CNT model, we treat the DMSO concentration as a constant, while the DMSO concentration profiles display noticeable drift along the microfluidic tube (see Supplementary Fig. 9d). Our further analysis shows that considering the DMSO concentration gradient does not have significant impact on the results (Supplementary Figs 22 and 23 and Supplementary Note 8).

**Free energy change of attaching a HPS monomer to an existing aggregate.** The free energy change associated with attaching a HPS monomer to an existing spherical aggregate containing $n$ molecules is

$$\Delta\Delta G_n = \gamma_{sl}\Delta A - h = kT\theta\left((n+1)^{2/3} - n^{2/3}\right) - h \quad (7)$$

where the first term $\gamma_{sl}\Delta A$ corresponds to the free energy cost associated with creating additional solute–solvent interface when the size of aggregate increases from $n$ to $(n+1)$. $\theta = \gamma_{sl}(36\pi)^{1/3}(v/p)^{2/3}/kT$. The second term, $h$, refers to the free energy of transferring a HPS molecule from the infinite-sized aggregate phase to the solution, where no additional interface between the aggregate and solution needs to be created. We followed ref. 61 to compute $h$:

$$h = -kT\ln\left(C_e/C_{agg}\right) \quad (8)$$

where $C_e$ is the solubility of HPS monomer in the solvent mixture and $C_{agg} = p/(\nu N_a) = 1.352$ M is the HPS molar concentration in aggregates. Here, $\nu$ is the molecular volume of a HPS monomer, $p$ is the packing density and $N_a$ is the Avogadro's constant. As the solute–solvent surface tension $\gamma_{sl}$ have already been obtained previously, we can then calculate the values of $\theta$ in various solvent mixtures (see Supplementary Table 2). As the solubility $C_e$ was measured in our experiments (see Supplementary Fig. 10 for details), we can also obtain the values of $h$ in various solvent mixtures (see Supplementary Table 2). Thus, free energy changes associated with attaching a HPS monomer to an existing aggregate in various DMSO/water solvent mixtures can be computed based on equation (7) and the results were shown in Fig. 3b.

We further determined the free energy associated with attaching a HPS monomer to aggregates in pure water (Fig. 1c). To achieve this, we obtained $h$ and solute–solvent surface tension $\gamma_{sl}$ in pure water as follows: first, by plotting the transferring free energy $h$ against the surface tension of solvent $\gamma_s$ at 20 °C (ref. 62; Supplementary Fig. 5a and Supplementary Table 1), we identified the relationship between $h$ and $\gamma_s$ to be linear ($R^2$ of 0.98). We then extrapolated this fitted linear curve (from solvent mixture condition) to obtain $h$ to be 13.6 kcal mol$^{-1}$ in pure water, yielding a HPS solubility of $9.6 \times 10^{-5}$ μM. Next, we plotted the solute–solvent surface tension of HPS aggregates $\gamma_{sl}$ against the surface tension of the solvent ($\gamma_s$) (Supplementary Fig. 5b), and a linear relationship was obtained ($R^2$ of 0.99). By extrapolating the fitted curve to the pure water condition, we obtained a solute–solvent surface tension of 31.8 cal mol$^{-1}$ Å$^{-2}$ for HPS aggregates in pure water. These two linear relationships ($h$ versus $\gamma_s$, and $\gamma_{sl}$ versus $\gamma_s$) were also shown in previous studies[63,64].

**Cooperativity in hydrophobic aggregation.** We followed Wang et al.[21] to define the cooperativity in hydrophobic interactions as the excess multibody free energy upon attaching a HPS monomer to an aggregate. For example, the cooperativity in a trimer formation is defined as the difference between the associated free energy to form a trimer and the summarized two-body potential of mean forces[20,22]. For systems containing more than three molecules, the free energy of an aggregate containing $n$ molecules, $W_n$, can be decomposed into the single-body term $\sum_i F_i$, the pairwise term $\sum_{i<j} \delta F_{ij}$ and the multibody term $\sum_i \delta F_i$:

$$W_n = \sum_i F_i + \sum_{i<j} \delta F_{ij} + \sum_i \delta F_i \qquad (9)$$

Then, the free energy of the aggregate containing $(n+1)$ molecules, $W_{n+1}$, can be decomposed into the single-body term $F_{n+1} + \sum_{i \leq n} F_i$, the pairwise term $\sum_{i<j;i,j \leq n} \delta F_{ij} + \sum_{i \leq n} \delta F_{i,n+1}$ and the multibody term $\sum_{i \leq n+1} \delta F_i = \sum_{i \leq n} \delta F_i + Q_n$, where $Q_n$ is the excess multibody term, which is the quantity of the cooperativity in our definition. Therefore, we have

$$W_{n+1} = F_{n+1} + \sum_{i \leq n} F_i + \sum_{i<j \leq n} \delta F_{ij} + \sum_{i \leq n} \delta F_{i,n+1} + \sum_{i \leq n} \delta F_i + Q_n \qquad (10)$$

After simple rearrangement, we obtained,

$$Q_n = W_{n+1} - W_n - F_{n+1} - \sum_{i \leq n} \delta F_{i,n+1} = \Delta\Delta G_n - \sum_{i \leq n} \delta F_{i,n+1} \qquad (11)$$

where $W_{n+1} - W_n - F_{n+1} = \Delta\Delta G_n$ is the free energy change associated with attaching a HPS monomer to an aggregate containing $n$ molecules, and $\sum_{i \leq n} \delta F_{i,n+1}$ is the summarization of the two-body potential of mean force between the $(n+1)^{th}$ molecule and the $i^{th}$ molecule in the aggregate. The cooperativity $Q_n$ defined by equation (11) can be zero (additive), negative (cooperative) or positive (anti-cooperative).

To compute the hydrophobic cooperativity as defined in equation (11), we need to obtain both $\Delta\Delta G_n$ and $\sum_{i \leq n} \delta F_{i,n+1}$. The first term has been obtained from our experiment (Figs 1c and 3b). To compute the second term, we need to consider the potential of mean force to bring together the $(n+1)^{th}$ molecule and the $i^{th}$ molecule in the aggregate from infinite distance away to a particular distance determined by the location of $i^{th}$ molecule in the aggregate. If the $(n+1)^{th}$ molecule is in contact with the $i^{th}$ molecule in the newly formed aggregate containing $(n+1)$ molecules, the magnitude of $\delta F_{i,n+1}$ is significant due to the hydrophobic interactions associated with the desolvation between these two molecules. However, if the $(n+1)^{th}$ molecule is separated from the $i^{th}$ molecule by other HPS molecules, their centre of mass distance is at least two nanometres ($\sim 2$ diameters of HPS). In this case, no desolvation is needed and $\delta F_{i,n+1}$ is thus negligible. Therefore, we only consider the contributions of these contact pairs to $\sum_{i \leq n} \delta F_{i,n+1}$, and then need to estimate the number of contact pairs in the aggregate. Bezdek and Reid[65] proved that the number of contact pairs in an arbitrary lattice packing of $n$ unit balls in three-dimensional Euclidian space is always greater than $(6n - 7.862n^{2/3})$, but less than $(6n - 3.665n^{2/3})$. To make the best estimation of contact pairs in a spherical aggregate, we proved that the number of contact pairs in the aggregate containing $n$ molecules is (see Supplementary Note 9 and Supplementary Fig. 24 for details):

$$P_n = 6n - 7.2n^{2/3} \qquad (12)$$

The estimated contact pairs are shown in Supplementary Fig. 12. Next, we need to obtain the two-body potential of mean force ($\delta F$) of a contact pair. $\delta F$ can be obtained from directly computing the free energy difference by bringing a pair of monomers from infinite distance to be in contact as suggested by Wang et al.[21] In our study, it is challenging to directly measure $\delta F$ in this way, because the smallest stable aggregate (that is the critical size plus one, according to the CNT) in experiment contains more than two molecules (see Supplementary Table 2 for details). Therefore, we estimated $\delta F$ by approximating it to the pair strength in the smallest stable aggregate in a given solvent:

$$\delta F = \sum_{k=1}^{n^*} \frac{\Delta\Delta G_k}{P_{n^*+1}} \qquad (13)$$

In particular, the smallest stable aggregate ($n^* + 1$) contains 3–7 molecules in the solvent with a DMSO mole fraction of 0, 0.16, 0.21, 0.26 and 0.32, respectively. $\Delta\Delta G$ can be obtained from equation (7). Thus, the summarization of the two-body potential of mean force between the $(n+1)^{th}$ molecule and the $i^{th}$ molecule in the aggregate is obtained as follows:

$$\sum_{i \leq n} \delta F_{i,n+1} = (P_{n+1} - P_n)\delta F = (P_{n+1} - P_n) \sum_{k=1}^{n^*} \frac{\Delta\Delta G_k}{P_{n^*+1}} \qquad (14)$$

Substituting equations (7) and (14) into equation (11), we determined the hydrophobic cooperativity (Figs 1c and 3b). We note that the data reported in Figs 1c and 3b,c start from the smallest stable aggregate in the corresponding experimental condition. As our reported hydrophobic free energies at large $n$ limit ($\Delta\Delta G_{n \to \infty}$) become equivalent to the free energy of transferring a HPS molecule from water to the HPS phase, this makes it possible to directly estimate the cooperativity even without performing microfluidics experiments, provided that one could also obtain the pair potential of mean force (for example, from MD simulations discussed in Supplementary Note 5 and Supplementary Figs 17 and 18).

**Kinetics of the hydrophobic aggregation.** To resolve the kinetics of HPS aggregation, we have tracked the growth of the first nucleus formed. Based on the theoretical model obtained previously, we derived the equation for nucleus growth as follows:

$$n(t) - n(t - \Delta t) = 4\pi DC_e \left[ \frac{3n(t - \Delta t)\nu}{4\pi p} \right]^{1/3} [S(t - \Delta t) - 1]\Delta t, n(0) = n^* \qquad (15)$$

where $n(t - \Delta t)$ is the size of the aggregate at time $t - \Delta t$. On the basis of equation (15), we obtained the nucleus growth curve as shown in Supplementary Fig. 7. Then, we defined the time for attaching a HPS monomer to the aggregate as the time required for the aggregate to grow by one molecule in size, and the results are shown in Figs 1d and 3c. The nucleation-growth curve (Supplementary Fig. 7) suggests the microsecond timescale of HPS aggregation, which is comparable to the timescale of early stage protein folding. This comparison is made between similar concentration of HPS in our experiment and local concentration of residues on a protein chain. We estimated local concentrations of protein residues by computing number of residues within a distance of the protein chain length (for example, a protein domain containing 89–100 residues has a chain length of 28–30 nm). The resulting local concentration of protein residues (1.3–1.8 mM) is comparable with our initial HPS concentration in the microfluidics tube: 2.0–6.0 mM.

**Data availability.** The data that support the findings of this study as well as the computer code for the CNT and non-CNT theories are available from the corresponding authors upon reasonable request.

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

## Acknowledgements

We thank Dr Lingle Wang for his critical reading of our manuscript. We acknowledge supports from National Basic Research Program of China (973 program, Grant No. 2013CB834703), Hong Kong Research Grants Council (Grant Nos. HKUST C6009-15G, AoE/P-705/16, ECS 60981, 621113, 16304215, F-HKUST605/15), and Innovation and Technology Commission (ITC-CNERC14SC01). We thank the Nanoelectronic Fabrication Facility, Biosciences Central Research Facility at HKUST and Dr Baoling Huang for providing technique support for microfluidics experiments.

## Author contributions

L.J. and S.Y. designed and performed the experiments. S.C., L.J. and X.H. established the classical nucleation and growth model for theoretical fitting and analysed fitting data. C.W.T.L. and B.Z.T. synthesized HPS and helped to perform solubility measurements. All authors discussed the results and contributed to the writing of the manuscript.

## Additional information

**Competing interests:** The authors declare no competing financial interests.

**Publisher's note**: 

