## [Peer Review File · Nature Communications]

Reviewers' comments:

Reviewer #1 (Remarks to the Author):

This manuscript presents results of nano-fluidic experiments aimed at quantifying the free energy and cooperativity associated with the aggregation of hexaphenylsilole (HPS) molecules in a supersaturated aqueous solution of HPS. The experiments are performed by rapidly mixing (on the microsecond time scale) solutions of HPS in DMSO with mixtures of water and DMSO (of various DMSO concentrations) and monitoring the HPS fluorescence intensity arising from the HPS aggregates. The free energies and cooperativities are obtained by fitting the measured aggregation rates to a model based on classical nucleation theory. Results pertaining to the aggregation of HPS in pure water are obtained by extrapolating the results to infinite DMSO dilution. I find this work to be quite novel and the subject matter is of sufficiently broad interest and importance to be appropriate for Nature Communications. More specifically, the experimental quantification of hydrophobic interaction free energies and cooperativities is important both for understanding biological self-assembly and for the design of self-assembled processes and devices. However, I have significant concerns regarding the accuracy of the quantitative results and conclusions of this work. Most importantly, the estimates of the pairwise potentials of mean force (ΔF) have been obtained in a way that is quite questionable, thus also raising into question the resulting magnitudes of the cooperative contribution to the aggregation free energy. Nevertheless, I find that the present results and analysis represent an important contribution to understanding hydrophobic interactions and their cooperativity. The following are my specific comments and suggestions.

1) The claim that "no experiment has been performed to quantify hydrophobic interactions in bulk environment" is not correct. See for example, the recently published review [Ben-Amotz, *Ann. Rev. Phys. Chem.* 67, 618 (2016)], and references therein.

2) The claim that hydrophobic interactions are a major driving force for protein folding has been brought into question by recent studies; see for example [Harris, *J. Phys.: Condens. Matter*, 28, 083003 (2016)], and references therein.

3) The present analysis relies heavily on classical nucleation theory; I wonder if that description is necessarily applicable to the nucleation of HPS; see for example [Gebauer, *Nano Today* 6, 564 (2011)], and references therein.

4) The solute-solvent interfacial surface tension values used in this work (see Table Supplementary Table 1) do not seem to be consistent with hydrophobic free energy per unit surface area (on page 3). More specifically, the latter values are significantly smaller than the former values, when converted to the same units. Also, note that the hydration free energy of n-alkanes in water is 5-10 kJ/mol, and are nearly chain-length independent, (see the review cited above).

5) I suggest that the authors use a self-consistent set of units, such as SI units throughout the manuscript (and Supplementary Information). For example, all energies could be expressed in kJ/mol, and surface tensions in kJ/mol per nm², rather than using kcal/mol for energies and dyne/cm for surface tensions, and cal/mol per Angstrom² for hydrophobic free energies.

6) The thermodynamic, transport, and hydrodynamic analyses presented in the SI are quite impressive. However, some of the results of these analyses have been pushed beyond their expected realm of applicability, particularly with regard to the estimated cooperativity (as further explained below).

7) The claimed cooperativity of the aggregation is based on a flawed argument. In particular, Eq. 14 in the SI is not expected to be applicable to aggregate sizes between 3-7, which are claimed to be the smallest stable aggregates of HPS in the aqueous DMSO solutions used in this work. The authors nevertheless use Eq. 14 to estimate the pair potential of mean force by applying Eq. 20 to the smallest stable aggregates. In other words, the resulting cooperativity is essentially an artifact arising from the application of Eq. 14 to very small aggregates. A better way to obtain the desired measure of cooperativity would be to perform a simulation of the mean force potential between two HPS molecules in each of the relevant aqueous DMSO solutions, as has been done, for example, for other large solutes in water [Makowski et al *JPC-B*, 114, 993 (2010)]. In particular, I suggest that the authors compare the values obtained using Eq. 20 with previously published simulation results for molecules of comparable size dissolved in water. I believe that they will find

that the magnitudes of the pair potentials of mean force obtained using Eq. 20 are not realistic. If, as I suspect may be the case, the true values of dF are smaller than those estimated using Eq. 20, and if Eq. 14 can be relied upon, then that implies that the actual cooperativity is larger than estimated using the analysis presented in the SI. Thus, I believe that the impact and significance of this work will increase if the authors do a better job of estimating the range of physically reasonable values of dF , and what that implies regarding the cooperativity of the hydrophobic aggregation of PHS.

8) In the large n limit the free energies shown in Figure 1c and 3b are simply equivalent to the transfer free energies of HPS from water to HPS (or more specifically the corresponding mutually equilibrated water-rich and HPS-rich phases). Thus, I think it could be misleading to refer to these as hydrophobic interaction free energies. I suggest that the authors make the above large n limit relation more clear. Note that this also implies that one can make quantitative inferences regarding the cooperativity of hydrophobic aggregation processes from comparisons of pairwise mean force potentials and macroscopic phase transfer free energies. That is an important point that the authors could make in this manuscript. Equation 19 and its very nice derivation will be very useful in that regard; I find it quite amazing that Eq. 19 produces reasonable predictions all the way down to $n=3$.

Reviewer #2 (Remarks to the Author):

The manuscript entitled "Real-Time Monitoring of Hydrophobic Aggregation Reveals a Critical Role of Cooperativity in Hydrophobic Effect" from Jiang et al. described an interesting experiment using microfluidic mixer to study the cooperativity of hydrophobic association using hexaphenylsilole (HPS) as a model system. In the microfluidic mixer, stream of HPS dissolved in DMSO was jetted and sandwiched between water layers, where DMSO quickly diffuses out and water diffuses into the HPS stream, initiating HPS aggregation. Upon aggregation, HPS becomes fluorescent, which the authors argues are linearly correlated to aggregate size. Under saturated condition, the authors were able to fit the fluorescence intensity profile along the flow to a classical nucleation growth model, where the solute-solvent surface tension can be obtained. This parameter allows the authors to compute other thermodynamic parameters such as critical aggregate size and free energy barrier. The authors found that the cooperative effect of hydrophobic interactions contribute to 40% of the total aggregation free energy in this model system. The results of the study are well supported by the rigorous experimental method and analysis. The supplementary information was sufficiently detailed for other researchers to reproduce the work.

The authors should provide a more detailed discussion of the cooperativity contribution to hydrophobic interaction in other biological systems to provide a better context of their result. Cooperativity in hydrophobic effect is well appreciated, especially in the protein folding. What is the significance of 40% cooperative in terms of protein folding, membrane assembly, and other self-assembly systems? The contribution due to cooperativity would depend on the size and chemical composition of the solute, as well as temperature and condition of the solvent. Therefore, 40% is specific to this model system. Hydrophobic interactions are length-scale dependent with a transition scale around 1 nm. HPS molecules are significantly larger (~ 1.2 nm) than individual amino acids (sub-nm), and their hydrophobic interactions are controlled by different mechanisms. Therefore, the implication of hydrophobic interaction in HPS aggregation to hydrophobic core formation in proteins may be limited. Although a very neat system, the experimental method relies on fluorescence due to aggregation, it is unclear how easily it can be applied to study hydrophobic interactions of other molecules and in other systems.

Line 61: The authors stated that this is the first bulk study of quantify hydrophobic effect. This is an overstatement – the majority of studies of hydrophobic effect is performed in bulk, besides a few single-molecule studies. The authors should clarify and be more specific what they mean here.

Line 106: Why is 40% a striking finding? What are other systems to compare with?

Line 136: The authors state that the aggregation time scale is comparable to those of early stage protein folding. This however, strongly depend on the solute concentration. The authors should comment on how the concentrations they used were "equivalent" to hydrophobic residues on a polymer chain for fair comparison.

Fig. 2b: It is unclear why the relative fluorescence of 3-6mM samples have non-zero y-intercepts. My understanding from the manuscript is that aggregation starts at time 0, and that no fluorescence should be observed before aggregation occurs, hence 0 fluorescence intensity at $t=0$. Are the authors simply offsetting the traces to make the graph clearer? If so, it should be clearly stated and better illustrated.

Fig. 2b: There is a weak dependency of the solute solvent surface tension from 14.5 to 14.2 on HPS concentration (6-2mM). Why does such dependency exist?

Supp.Fig.4: One of the key assumptions is that the particle volume scales linearly to fluorescence, regardless of the size of the aggregate. The evidence provided is a linear plot of fluorescence intensity vs particle size from AFM measurement. In this experiment, the particle volume is on the order of 10^6 nm^3 , or $100 \times 100 \times 100 \text{ nm}$ (~ 1 million HPS molecules), these are huge aggregates and not surprising that total fluorescence is proportional to the volume. At this size scale, it is easy to extrapolate a fit close to 0. The size of aggregates formed in the flow are much smaller - between 2-10nm, or 5-100 molecules. Therefore, if the line does not intercept exactly at 0, the fluorescence-size linearity would fail for small aggregates. It is not immediately clear how the fluorescence of very small aggregates (2, 3, 4, 5 HPS molecules) are necessarily linearly correlated to the number of HPS in the aggregate.

Equation 1: It wasn't clear how the authors took into account the DMSO concentration gradient along the jet in their nucleation growth model. Additionally, the DMSO concentration profile along the jet was calculated but not experimentally verified (Supp.Fig.9d). The authors should comment on the reliability of such calculation. One can imagine using a dye in the same experimental setup to validate such simulation.

Lastly, it would be nice if the authors could show how the cooperativity of hydrophobic contribution is dependent on temperature and isolate the entropic vs enthalpic components.

Reviewer #3 (Remarks to the Author):

In "Real-Time Monitoring of Hydrophobic Aggregation Reveals a Critical Role of Cooperativity in Hydrophobic Effect" the authors present a micro-fluidic study of the mixing of HPS/DMSO solutions in water to examine the kinetics and thermodynamics of the hydrophobic aggregation of HPS in water. First I would like to say that this appears to be a unique paper that has a lot of interesting data in it. I believe the results from this paper would be of great interest to researchers interested in hydrophobic interactions and assembly. The results would also be a provide data of interest to simulation and theoretical scientists interested in modeling hydrophobic effects. On that basis I believe this paper would find an audience in Nature Communications. I hope that this work is followed up by a longer paper with more details on the techniques and more data would be published in an archival journal, however, to provide more results for scientists to be able to follow up on. Despite the novelty of this contribution I do have problems with the presentation of this work that should be fixed before it is published. I enumerate my concerns below:

- In the abstract and in the introduction the authors seem to imply that hydrophobic interactions

are 'fundamental' like ionic, dipolar, and dispersion forces. This is simply not true. The hydrophobic interaction is at its heart a thermodynamic force that arises after non-polar moieties are placed in an aqueous medium. Hydrophobic interactions do not exist outside of this context. Ionic, dipolar, and dispersion interactions, on the other hand, exist in all contexts in which there are molecules involved, arising from the arrangements of electrons about atomic nuclei. In this case then, I would consider ionic, dipolar, and dispersion interactions to be fundamental, while hydrophobic interactions. Rather hydrophobic interactions arise from the contortions water has to take about non-polar species in order to accommodate non-polar solutes.

The authors imply in the introduction that only hydrophobic interactions are cooperative, while the others are only pairwise additive. This is approximately true, but there are significant efforts trying to incorporate polarizability and multi-body effects into simulations, which are clearly not pairwise additive. Moreover, pairwise additive interactions in protein folding can also give rise to cooperative effects. The most prominent example of this is the formation of protein helices by hydrogen-bonding down the peptide backbone. Helix melting curves point to a cooperative nature, which has been successfully captured by the Zimm-Bragg and Lifson-Roig helix-coil models which treat hydrogen-bonding as simply pairwise additive. It would be hard in my mind to try to disentangle the cooperativity the authors investigate from other sources of cooperativity in a protein folding experiment. This work does not effect that. I would also think that part of the cooperativity in proteins folding arises from the spatial locality of the aggregating units along the peptide backbone, not dispersed molecules coming together to form an aggregate. In this case the time scales for protein folding and the present experiments would not be comparable.

- There is no indication in the introduction what the authors are going to do to place how their contribution would fit into the wider knowledge base on hydrophobic interactions. Rather than authors jump into the description of what their contribution is starting at the beginning of the results and discussion section. While I am not an expert on microfluidic mixing, surely this has been done before. It seems like it would be wise in the introduction discuss how this technique applies to the problem at hand. The way this reads now is just jarring.
- The authors do point to previous theoretical work on large scale hydrophobic interactions, but they do not take steps to try to connect their work to those efforts. For example, Lum, Chandler, and Weeks (ref 3 in this work) make the point that the differences between molecular-scale versus meso-scale arise from a crossover from volume based hydration thermodynamics for surfaces less than 1 nm in radius versus area based thermodynamics for larger scale solutes. This crossover appears as a kink when they plot the free energy of forming a solute of a given size on a per surface area basis versus the radius of the solute. In the case of aggregates this suggests that a kink would arise if the free energy for forming a cluster divided by the aggregation number raised to the 2/3's power ($N^{2/3}$ is proportional to the aggregate surface area) was plotted against $N^{1/3}$ (proportional to the aggregate radius). Cooperativity in the LCW theory then arises from the differences in scaling of the free energy on one side of the kink versus the other. It would be instructive if the authors attempted this type of calculation to touch base with existing theoretical/scaling ideas associated with hydrophobic interactions.

I do wonder if the cooperativity described here is even surprising. As the aggregates get bigger and bigger they are effectively form a pure HPS phase. Phase separation is certainly a cooperatively process because there is no new phase unless a lot of things came together to make that phase. When I look at the data in Figure 1c then I find myself wondering if the authors have simply just determined the free energy of transferring a single HPS into its neat liquid (? - or solid). Is this known from independent measurements? I know Walker and Li demonstrated the correlation between their pulling experiments with solubility measurements - has any such correlations been attempted here?

Indeed the cooperativity of micelle formation has long been associated with the free energy of transferring surfactant hydrocarbon tails into a neat liquid phase - the so-called phase separation model of micellization. I can't tell if the authors have adding anything more beyond this model for assembly, which is quite old and accurate.

- The authors seem to want to compare their aggregation free energies against the energy of water forming a hydrogen-bond. I realize this comes about because of the cartoon picture put forward in previous work that the formation of large surfaces is accompanied by the loss of hydrogen bonds. Water would already not be able to form a complete hydrogen-bonding network about a single HPS model with its molecular detail, nooks and crannies. Aggregation in this case is not likely to change the extent of water hydrogen bonding as the aggregates get bigger and bigger. Rather this is likely a surface tension driven effect. The crossover from volume to surface area based interactions in the LCW theory is not simply a result of hydrogen bonding. Indeed LCW theory has no hydrogen bonding in it. Rather the crossover occurs as a result of density fluctuations changing from approximately Gaussian in nature for small scale solutes, to distinctly non-Gaussian for larger aggregates as a result of the thermodynamic proximity of the solvent (water in this case) to a liquid-vapor phase transition. The solvent then does not have to have hydrogen-bonding, but simply be close to a phase transition. Water is unique amongst solvents, however, do to its larger surface tension (which may be argued arises from hydrogen bonding). Nevertheless, the cooperativity the authors report likely is not a result of hydrogen-bonding but the closeness to water's vaporization transition.

- Beginning on line 120 the authors state "As hydrophobic interaction is a major driving force for protein folding, proteins will be likely unable to fold into their three-dimensional structures in the absence of hydrophobic cooperativity." This statement cannot be concluded from the HPS aggregation measurements performed herein. Indeed folded protein conformations are stabilized by energies on the order of 10 kcal/mol. The authors find aggregate formation free energies much larger than this for aggregates as small as 10 HPS's (see line 97). The free energies for stabilizing proteins thus are considerable smaller than what they report in the cooperative regime for larger aggregates. Cooperative boosts to the folding free energy do not seem to be necessary for protein folding based on the authors own results. Moreover, hydrophobic interactions are not thought to stabilize the three-dimensional structure of proteins. Rather these interactions are thought to help collapse the chain into a globular shape. Three dimensional secondary structure formation however certainly need hydrogen bonding and side chain packing to arrive at its final shape.

- Line 143. The authors write "nature designs hydrophobic interaction" is not true. Nature doesn't design the hydrophobic interaction. It might make use of it and take advantage of it to its own ends, but it doesn't design it.

- Line 145. The authors write "Without cooperativity, proteins may not be able to fold to specific structures, and as a consequence all life formed on Earth may perish." My jaw just dropped when I read this concluding sentence. This sentence reads like we need to do something to make sure hydrophobic interactions remain cooperative or we are all going to die. This is a bit of a grandiose misstatement. We are not in danger of losing cooperativity. Without the ability for proteins to fold, the more appropriate statement would be that life as we know it would not have appeared. Since we are here, however, it looks like the hydrophobic effect is working just fine and we are not imperiled. This is a gross oversell.

- The whole concluding paragraph on page 4 has almost nothing to do with the rest of the paper. We already knew that hydrophobic aggregation was faster than the final protein folding, so if the only conclusion of this paper is something we already knew. There is no summary of the salient conclusions that can be drawn from this work and the extrapolations are grandiose and do not directly follow from the rest of the paper. This whole paragraph should be written in a thoughtful manner.

- The paper is rife with grammatical errors, awkward wordings, and throw away sentences that challenge reading of this work. For instance:

- Line 12. Insert a "The" before "hydrophobic" in the first sentence of the abstract. Also the hydrophobic interaction is not fundamental (see above).

- Line 34. Awkward wording "... of hydrophobic effect ...". Sentence needs to be rewritten.

- Line 48. "... and insofar focused ...". Insofar means "to the extent that" so this statement makes no sense.
- Line 120. "As hydrophobic interaction is a major ...". Awkward sentence beginning. Not grammatical.
- Line 123. "... at tenths to hundredths of nanoseconds." I think they mean tens to hundreds. Tenths and hundredths of nanoseconds are 1/10 and 1/100 nanoseconds.
- Line 129. "Slows" not "slow"
- Line 138. I know biochemists might call lipids macromolecules, but they are not.
- Line 143. "the hydrophobic interaction" not "hydrophobic interaction"

All in all I think there is something here, but in its present form I cannot support publication. The final conclusions drawn don't seem warranted and over hyped. I would certainly site this work, but I would prefer a longer publication with more details.

Response to Reviewer #1's comments

We appreciate this reviewer's recognition of the novelty, broad interest and importance of our work. In particular, he/she thinks that our results represent an important contribution to the understanding of hydrophobic interactions and their cooperativity. However, this reviewer also raised several major concerns, particularly on the estimation of pair potential of mean force to further improve our manuscript.

- 1. The claim that "no experiment has been performed to quantify hydrophobic interactions in bulk environment" is not correct. See for example, the recently published review [Ben-Amotz, *Ann. Rev. Phys. Chem.* 67, 618 (2016)], and references therein.**

Reply: We thank the reviewer for this good comment and pointing out to us a highly relevant review article. We agree with the reviewer that earlier experimental attempts (e.g. Ref 88, 96-104 in *Ann. Rev. Phys. Chem.* 67, 618 (2016)) have been made to estimate hydrophobic interactions particularly to assess the potential of mean force in bulk solution. In this revised manuscript, we have revised the sentence as suggested by the reviewer as follows:

"...Although earlier experimental³³ studies have attempted to estimate water-mediated hydrophobic interactions, it remains challenging to directly monitor the process of hydrophobic aggregation in bulk environment and further quantify hydrophobic interactions..."

- 2. The claim that hydrophobic interactions are a major driving force for protein folding has been brought into question by recent studies; see for example [Harris, *J. Phys.: Condens. Matter*, 28, 083003 (2016)], and references therein.**

Reply: We thank the reviewer for this comment. We have revised manuscript to remove this sentence, and included the reference pointed out by the review in the discussion section of the maintext:

"... Organic molecules, such as proteins and lipid, bury their hydrophobic components to form stable cores. Hydrophobic interaction underlines its crucial role in facilitating the collapse of protein chains into a globular shape^{25, 52-54}"

- 3. The present analysis relies heavily on classical nucleation theory; I wonder if that description is necessarily applicable to the nucleation of HPS; see for example [Gebauer, *Nano Today* 6, 564 (2011)], and references therein.**

Reply: We would like to thank the reviewer for this great comment. We agree with the reviewer that our analysis heavily relies on the classical nucleation theory (CNT), and thus we have followed the reviewer's suggestion to apply an alternative model to our analysis: a non-classical nucleation theory (non-CNT) (*Nano Today* 6, 564 (2011)). In non-CNT, the existence of stable prenucleation clusters will introduce a second free energy barrier, ε (see Eq. R1), in comparison to CNT containing only a single barrier. Applying non-CNT to our system, we show that ε has to be within $1.2 kT$ (comparable with thermal fluctuations) in order to obtain reasonable agreement with experiment (see Fig. R1-2). Not surprisingly, with these small values of ε , the non-CNT theory provides consistent results with our CNT theory in both $\Delta\Delta G$ and cooperativity

(see Fig. R3). These results suggest that the formation of stable prenucleation clusters is not substantial in the aggregation of HPS molecules, and thus we believe that CNT is applicable to our system. In literature, the non-CNT theory is found to be particularly useful when describing systems such as crystallization of ionic compounds and proteins (*Nano Today* 6, 564 (2011)). We anticipate that hydrophobic interactions that drive HPS aggregation is non-specific in nature, and this is in contrast to the formation of specific ion pairs during the crystallization of ionic compounds. Therefore, the HPS aggregation process may not favor the formation of prenucleation clusters as much as ionic compounds. In addition, as HPS is more rigid compared to flexible polymers like proteins, a second free energy barrier due to the re-arrangement of protein upon initial collapse is not likely to be present in our HPS system. Please see below for the detailed analysis of the non-CNT theory. Once again, we thank the reviewer for this great suggestion!

We constructed the non-CNT theory based on the Dillmann-Meler model (*PRL* 108, 225701 (2012)) that involves two free energy barriers:

$$\Delta G(n) = \theta n^{2/3} - nh + \varepsilon \quad (\text{R1})$$

Where the first two terms are identical with CNT (see SI Eq. S14 for details), while ε corresponds to the effective free energy barrier for the transition from nucleated amorphous clusters to the crystalline phase per molecule (see Fig 1 in *Science* 322, 1802 (2008)). This second free energy barrier ε will introduce corrections on both nucleation rate (J) and growth rate (f) (*JCP* 93, 1273, (1990)):

$$J_{\text{non-CNT}} = e^{-\varepsilon} J_{\text{CNT}} \quad (\text{R2})$$

$$f_{\text{non-CNT}} = e^{-\varepsilon} f_{\text{CNT}} \quad (\text{R3})$$

When ε equals 0, the non-CNT theory is reduced to the CNT theory. When $\varepsilon \gg kT$, the second free energy barrier becomes the rate-limiting step. Please refer to SI Sec. 7 for more details of the non-CNT theory.

We show that ε has to be within $1.2 kT$ in order to obtain reasonable fitting to experimental data. After applying nucleation and growth rates defined in Eq. R2-3 to our model, we examined the quality of model fitting to experimental fluorescence over a wide range of ε (0, 0.1, 0.5, 1.2, 2.3, 3.0, 3.9, and 4.6 kT). As shown in Fig. R1, when $\varepsilon > 1.2kT$, the non-CNT model provides very slow aggregation rates and the final aggregate size is still within 15. As suggested in Ref 45 (*Nanoscale*, 8, 15173 (2016)), aggregates with this small size will have majority of the HPS molecules exposed to solvent with low quantum yields, and thus may not be able to emit detectable fluorescence. Consistently, we notice that the relative deviations of theoretical fitting from experiment (on the average slope of the total fluorescence intensity) become more significant with the increase of ε among all HPS initial concentrations (see Fig. R2). Therefore, ε has to be within $1.2kT$ in order to obtain reasonable agreement between the non-CNT theory and experiment.

Fig. R1. (a). The averaged aggregate size at the end of the microfluidic tube ($t = 150 \mu s$) obtained from the non-CNT model with various values of ϵ ($\epsilon = 0, 0.1, 0.5, 1.2, 2.3, 3.0, 3.9$, and $4.6 kT$) under experimental conditions with different initial HPS concentrations ($[HPS] = 6, 4, 3$, and 2 mM). When $\epsilon = 0 kT$, the non-CNT model is equivalent to a CNT model. (b). The same as (a) except that the average aggregate size as a function of time is displayed.

Fig. R2. The deviation of theoretical fitting from experiment on the slope of the fluorescence curve: $\delta = \langle (\dot{i}_{exp}(t) - \dot{i}_{theory}(t)) / \dot{i}_{exp}(t) \rangle$. The non-CNT model was applied for the theoretical fitting with various values of ε ($\varepsilon = 0, 0.1, 0.5, 1.2, 2.3, 3.0, 3.9,$ and $4.6 kT$) under experimental conditions with different initial HPS concentrations ($[HPS] = 6, 4, 3,$ and 2 mM). Specifically, segments of fluorescence curves, $0.5 < I < 0.95$, which cover major parts of the aggregate growth were taken into account to calculate the deviation.

When $\varepsilon \leq 1.2kT$, the second free energy barrier is at the order of thermal fluctuations or even smaller, which should not have substantial impact on the nucleation process. Indeed, we show that the predictions of non-CNT theory match well with those from the CNT theory in both $\Delta\Delta G$ and cooperativity for different systems (see Fig. R3). Therefore, we conclude that the CNT theory can sufficiently well describe the HPS aggregation process in this study.

Fig. R3. Hydrophobic free energy ($\Delta\Delta G$) and cooperativity computed from the non-CNT model with different values of ε . (a) $\Delta\Delta G$ of HPS aggregation in pure water when ε is 0 (black, reduced to the CNT model), 0.1 (red), 0.5 (magenta) and 1.2 (blue), respectively. (b) Cooperativity of attaching a HPS molecule to an infinite sized HPS aggregate, where the black box, grey, magenta, green and yellow bars represent the systems in pure water, DMSO/water solvent mixtures with 16%, 21%, 26% and 32% of DMSO, respectively.

To include the above discussions on the non-CNT model, we have added the following sentences in the maintext of the revised manuscript:

“...As our quantitative analysis heavily relies on the CNT, we have examined an alternative theory: a non-classical nucleation theory (non-CNT) involving stable prenucleation clusters (see Supplementary Sec. 7 for details). In non-CNT, the existence of stable prenucleation clusters will introduce a second free energy barrier, ε (Eq. S23), in comparison to CNT containing only a single barrier. Applying non-CNT to our system, we show that ε has to be within 1.2 kT (comparable with thermal fluctuations) in order to obtain reasonable fitting to experimental fluorescence (Supplementary Fig. 14-15). Not surprisingly, with these small values of ε , the non-CNT theory provides consistent results with our CNT theory in both $\Delta\Delta G$ and cooperativity (Supplementary Fig. 16). These results suggest that the formation of stable prenucleation clusters is not substantial in the aggregation of HPS molecules, and thus CNT is well applicable to our system....”

In addition, we have added the details of the non-CNT theory in the revised manuscript as SI Sec. 7: “The non-classical nucleation and growth theory”. Finally, we have included Fig. R1, R2, R3 in the revised manuscript as SI Fig. S14, S15, and S16, respectively.

4. The solute-solvent interfacial surface tension values used in this work (see Table Supplementary Table 1) do not seem to be consistent with hydrophobic free energy per unit surface area (on page 3). More specifically, the latter values are significantly smaller than the former values, when converted to the same units. Also, note that the hydration free energy of n-alkanes in water 5-10 kJ/mol, and are nearly chain-length independent, (see the review cited above).

Reply: The surface tension values listed in Supplementary Table 1 are actually solvent-vapor surface tensions (γ_{lv}), while the solute-solvent surface tension (γ_{sl}) values are displayed in Supplementary Table 2. In pure water solution, the hydrophobic free energy per unit surface area reported in Page 3 (18 cal/mol/Å²) is consistent with the solute-solvent surface tension (22.1 dyn/cm) listed in Supplementary Table 2 after unit conversion. To avoid confusion, we have adopted the same unit system for all reported values in the revised manuscript. In addition, in the caption of Supplementary Table 1, we have explicitly stated that the reported values correspond to solvent-vapor surface tensions.

The reviewer also pointed out that the hydration free energy of n-alkanes remains nearly invariant with the increase of chain length, while in our work the hydrophobic free energy continuously decreases with the aggregate size. We anticipate that this difference may be due to the entropic contributions. As flexible alkane chains contain numerous conformations, entropic contributions are positive and readily increase with the chain length, and thus largely compensate the decrease of enthalpy, thereby resulting in nearly constant free energy (see Fig. 2 in **Ann. Rev. Phys. Chem.** 67, 618 (2016)). In our study, as HPS molecules are small and relatively rigid, we anticipate that their intra-molecular entropic contributions to free energy are much less significant.

5. I suggest that the authors use a self-consistent set of unit, such as SI units throughout the manuscript (and Supplementary Information). For example, all energies could be expressed in kJ/mol, and surface tensions in kJ/mol per nm², rather than using

kcal/mol for energies and dyne/cm for surface tensions, and cal/mol per Angstrom² for hydrophobic free energies.

Reply: We would like to thank the reviewer for this good suggestion. In the revised manuscript, we have adopted a self-consistent set of unit throughout the maintext and SI: Energy in kcal/mol, distance in Å, time in μ s, wavelength in nm, and surface tension in cal/mol/Å².

- 6. The claimed cooperativity of the aggregation is based on a flawed argument. In particular, Eq. 14 in the SI is not expected to be applicable to aggregate sizes between 3-7, which are claimed to be the smallest stable aggregates of HPS in the aqueous DMSO solutions used in this work. The authors nevertheless use Eq. 14 to estimate the pair potential of mean force by applying Eq. 20 to the smallest stable aggregates. In other words, the resulting cooperativity is essentially an artifact arising from the application of Eq. 14 to very small aggregates. A better way to obtain the desired measure of cooperativity would be to perform a simulation of the mean force potential between two HPS molecules in each of the relevant aqueous DMSO solutions, as has been done, for example, for other large solutes in water [Makowski et al JPC-B, 114, 993 (2010)]. In particular, I suggest that the authors compare the values obtained using Eq. 20 with previously published simulation results for molecules of comparable size dissolved in water. I believe that they will find that the magnitudes of the pair potentials of mean force obtained using Eq. 20 are not realistic. If, as I suspect may be the case, the true values of dF are smaller than those estimated using Eq. 20, and if Eq. 14 can be relied upon, then that implies that the actual cooperativity is larger than estimated using the analysis presented in the SI. Thus, I believe that the impact and significance of this work will increase if the authors do a better job of estimating the range of physically reasonable values of dF , and what that implies regarding the cooperativity of the hydrophobic aggregation of HPS.**

Reply: We totally agree with the reviewer that accurately estimating the pair potential of mean force (dF) is important for the measure of cooperativity in our study. As it is challenging to directly measure dF from experiment, we estimated dF as averaged pair interactions in smallest stable aggregates (size 3-7) using SI Eq. 14 from the CNT theory. To examine the potential bias of SI Eq. 14 on resulting cooperativity, we followed the reviewer's suggestion to adopt the computational approach to directly compute dF . Since the HPS molecule has distinct chemical structure and size from other compounds reported previously (Makowski et al *JPCB*, 114, 993 (2010)), we decided to perform molecular dynamics (MD) simulations with umbrella sampling to compute the potential of mean force (PMF) for bringing together a pair of HPS molecules in water solution (~1100-ns simulations in total, see Fig. R4). From MD simulations, we obtained $dF = -1.30 \pm 0.15$ kcal/mol per molecular contact. As speculated by this reviewer, the mean value of dF is indeed slightly weaker than our previous estimation based on SI Eq. 14 (-1.33 kcal/mol), even though these two values are well within the uncertainty of computation. Subsequently, we show that the cooperativity is also slightly increased by ~2% if we adopt the dF value estimated from MD simulations. These results suggest that dF computed from the CNT theory (SI Eq. 14) and MD simulations are in reasonable agreement in predicting

cooperativity associated with the HPS aggregation. We are grateful for this great suggestion from the reviewer. Please see below for the details of our umbrella sampling calculations.

To compute the potential of mean force between a pair of HPS molecules, we adopted the umbrella sampling technique with all-atom MD simulations in the explicit solvent. In umbrella sampling, we have applied harmonic potentials to restrain the pair of HPS molecules to be at specific distances (i.e. 5, 6, 7, ... 20 Å in *Si-Si* distance), and performed 20 or 40-ns MD simulations at each distance window. We then applied the Weighted Histogram Analysis Method (WHAM) to remove bias due to these restraint potentials, and obtained the PMF curves (see SI Sec. 8 for simulation details). In addition, we noticed that all-atom representations of HPS in our MD simulations is not perfectly spherical; as such, this will subsequently affect its packing in HPS aggregates. In particular, each contact of two HPS molecules in MD simulations contains different number of contacting atom pairs (e.g. averaged number of atomic contacts per molecular contact in simulations of HPS dimer, aggregates of size 10, 40, 50 and 60 are 22.4, 9.1, 7.7, 7.5 and 7.2 respectively, MD simulations of aggregates are taken from *Nanoscale*, 8, 15173 (2016)). From the dimer PMF curves, we can estimate the average strength of individual atomic contact (by dividing 22.4), and further obtain effective PMF per molecular contact in aggregates (e.g. aggregates of size 60 contains 7.2 atomic contacts per molecular contact). As the averaged atomic contact number remain nearly invariant for aggregates of size larger than 40, we applied an atomic contacts/molecular contact of 7.2 to the PMF curves reported in Fig. R4. To ensure the convergence of conformational sampling, we have performed two independent sets of simulations: one was bringing two HPS molecules together from 20 to 5 Å, while the other one was separating two molecules apart from 5 to 20 Å. As shown in Fig. R4, these two independent sets of simulations generated PMF curves that were in reasonable agreement (see Fig. R5). To extract the strength of pair interactions from the PMF curves, we computed Boltzmann weighted average of PMF values in the first free energy minimum ($6.4\text{\AA} \leq d_{Si-Si} \leq 8.4\text{\AA}$).

Fig. R4. Potential of mean force (PMF) per molecular contact of bringing two HPS molecules together in water solution obtained from umbrella sampling molecular dynamics (MD) simulations. PMF curves from two independent sets of simulations were reported: one was bringing two HPS molecules together from 20 to 5 Å in *Si-Si* distance (in blue), while the other one was separating two molecules apart from 5 to 20 Å (in red). The window width of the umbrella sampling is chosen to be 1 Å, and the restraint force constant was set to be 4.78×10^4 kcal/mol/Å². Please refer to SI Sec. 8 for details of the simulation setup.

Based on the PMF curves, we obtained $dF = -1.30 \pm 0.15$ kcal/mol per molecular contact. This value suggests a slightly weaker pair interaction compared with our previous estimation based on the smallest stable aggregates using the CNT theory ($dF = -1.33$ kcal/mol). Nevertheless, the two estimations are within the uncertainty of computation. As shown in Fig. R6, the resulting cooperativity also increases slightly by $\sim 2\%$, even though the CNT theory prediction is well within the standard deviation of MD simulation results. We also note that the free energy calculations based on MD simulations and force fields often introduce errors of around 0.5 to 1 kcal/mol even for the hydration of small chemical compounds (*J Comput Aided Mol Des.*, 28:711, (2014)). Therefore, we decided to continue with our original model for dF calculations in the revised manuscript, and include MD simulation results as an alternative way of obtaining dF in the SI.

Fig. R5. The contribution of cooperativity to hydrophobic free energy as a function of aggregate size was displayed. Results from the CNT theory with pair potential of mean force (dF) estimated by SI Eq. 14 were plotted in red, while results with dF and its uncertainty computed from umbrella sampling MD simulations were displayed in blue.

To include the above discussions on the calculations of pair PMF and MD simulation results, we have a new section in the revised manuscript as SI Sec. 8 “An alternative way of estimating pair potential of mean force by umbrella sampling molecular dynamics simulations”, and refer to it in the maintext. We have also included Fig. R4 and R5 in the revised manuscript as Fig. S17 and S18, respectively.

- In the large n limit the free energies shown in Figure 1c and 3b are simply equivalent to the transfer free energies of HPS from water to HPS (or more specifically the corresponding mutually equilibrated water-rich and HPS-rich phases). Thus, I think it could be misleading to refer to these as hydrophobic interaction free energies. I suggest that the authors make the above large n limit relation more clear. Note that this also implies that one can make quantitative inferences regarding the cooperativity of**

hydrophobic aggregation processes from comparisons of pairwise mean force potentials and macroscopic phase transfer free energies. That is an important point that the authors could make in this manuscript.

Reply: We would like to thank the reviewer for this great suggestion. We totally agree with the reviewer's proposition that the reported hydrophobic free energies at large n limit ($\lim_{n \rightarrow \infty} \Delta\Delta G$) become equivalent to the transfer free energy of a HPS molecule from water to the HPS phase. This makes possible the estimation of the hydrophobic cooperativity without performing any microfluidics experiments, provided that one could also obtain the pair potential of mean force. This is a very interesting point, and we have added the following sentences in the revised manuscript to discuss it:

"...Interestingly, our reported hydrophobic free energies at large n limit ($\lim_{n \rightarrow \infty} \Delta\Delta G = -13.6$ kcal/mol) become equivalent to free energy of transferring a HPS molecule from water to the HPS phase..." in maintext, and "... As our reported hydrophobic free energies at large n limit ($\lim_{n \rightarrow \infty} \Delta\Delta G$) become equivalent to the free energy of transferring a HPS molecule from water to the HPS phase, this makes it possible to directly estimate the cooperativity even without performing microfluidics experiments, provided that one could also obtain the pair potential of mean force (e.g. from MD simulations as discussed in Supplementary Sec. 8)." in SI Sec. 8.

Response to Review #2

This reviewer thinks that we reported interesting results on cooperativity of hydrophobic associations of HPS molecules, which are well supported by rigorous experimental method and analysis. In the meanwhile, he/she also raised a few comments for us to further improve our manuscript.

- 1. The authors should provide a more detailed discussion of the cooperativity contribution to hydrophobic interaction in other biological systems to provide a better context of their result. Cooperativity in hydrophobic effect is well appreciated, especially in the protein folding. What is the significance of 40% cooperative in terms of protein folding, membrane assembly, and other self-assembly systems? The contribution due to cooperativity would depend on the size and chemical composition of the solute, as well as temperature and condition of the solvent. Therefore, 40% is specific to this model system. Hydrophobic interactions are length-scale dependent with a transition scale around 1 nm. HPS molecules are significantly larger (~1.2nm) than individual amino acids (sub-nm), and their hydrophobic interactions are controlled by different mechanisms. Therefore, the implication of hydrophobic interaction in HPS aggregation to hydrophobic core formation in proteins may be limited.**

Line 106: Why is 40% a striking finding? What are other systems to compare with?

Reply: We would like to thank the reviewer for this great comment. In our study, ~40% of cooperativity is measured for the hydrophobic association of HPS molecules in water solution. As the formation of hydrophobic cores of proteins also involves association of hydrophobic side-chains, we anticipate that our results have implications in protein folding as most proteins are stabilized by only a few kcal/mol in free energy (e.g. *Science*, 276, 1109, (1997)). We also agree with the reviewer that the aggregation of dispersed HPS molecules is substantially different from folding of flexible protein chains. In particular, these two processes are associated with different entropic contributions because flexible protein chains contain numerous conformations, while our HPS molecules are small and relatively rigid. Furthermore, protein folding is a complicated process that may involve different sources of cooperativity. For example, intra-chain hydrogen bonding can lead to a cooperative helix melting process (*JACS*, 131: 2306, (2009)). In addition, even though individual amino acid is smaller than a HPS molecule, many amino acid side-chains will be brought together in spatial proximity upon the formation of protein core, which leads to a size larger than the cross-over length scale for hydrophobic interactions at ~1nm. Finally, we agree with the reviewer that the contributions due to cooperativity would depend on many factors such as solute hydrophobicity, its chemical structure, solvent condition, as well as the temperature. Nevertheless, we believe that our reported cooperativity has implications on part of cooperativity in globular protein folding that arises from the formation of hydrophobic cores, as both processes are underlined by water-mediated hydrophobic interactions.

To include the above discussions, we have included the following sentences in the maintext of the revised manuscript:

“... Most importantly, we show that hydrophobic interaction, which is an interaction induced by collective behaviors of many water molecules, is strongly cooperative, and thus substantially enhance its strength during the aggregation of dispersed hydrophobic molecules in solution. We anticipate that our findings have profound implications in protein folding, as the protein core formation involves the collapse of hydrophobic side-chains. We acknowledge that these two processes are different in many aspects. For instance, flexible protein chains contain numerous conformations, while HPS molecules are relatively rigid. In addition, multiple sources may contribute to cooperativity in protein folding, such as the cooperative helix melting process due to hydrogen bonding⁵⁵. In spite of these differences, our findings highlight the important role of hydrophobic cooperativity (as large as 40%) in the initial collapse of protein chain to form into a globular shape. We expect that our experimental platform will have promising applications in studying hydrophobic interactions of a wide range of organic molecules with aggregation-induced emission⁵⁶, and in investigating the impact of important factors such as temperature on hydrophobic effect....”

In addition, we have removed the word “Strikingly” in Line 106 to avoid confusion.

2. Although a very neat system, the experimental method relies on fluorescence due to aggregation, it is unclear how easily it can be applied to study hydrophobic interactions of other molecules and in other systems.

Reply: We agree with the reviewer that our experimental method relies on solute molecules with characteristics of fluorescence emission induced by aggregation. In the literature, a wide range of such aggregation-induced-emission molecules has been reported (see *Chem. Rev.* 115, 11718, (2015)), which could potentially be suitable for our experimental platform to investigate hydrophobic aggregations. In addition, other fluorescence probes could also be coupled with microfluidics experiment to study conformational dynamics including folding and aggregation (e.g. *Biophys. J.* 93, 218 (2007)).

To include the above discussions, we have added the following sentence in the maintext:

“... We expect that our experimental platform will have promising applications in studying hydrophobic interactions of a wide range of organic molecules with aggregation-induced emission⁵⁶ ...”

3. Line 61: The authors stated that this is the first bulk study of quantify hydrophobic effect. This is an overstatement – the majority of studies of hydrophobic effect is performed in bulk, besides a few single-molecule studies. The authors should clarify and be more specific what they mean here.

Reply: We thank the reviewer for this great comment. As pointed out also by Reviewer #1, there many experimental attempts has been reported to estimate hydrophobic interactions, particularly the potential of mean force in bulk solution (e.g. Ref 88, 96-104 in *Ann. Rev. Phys. Chem.* 67, 618 (2016)), even though these studies did not directly monitor the hydrophobic aggregation process. We apologize for the inaccurate expression of our sentence. In the revised manuscript, we have revised the sentence commented by the reviewer as follows:

“...Although earlier experimental studies have attempted to estimate water-mediated hydrophobic interactions, it remains challenging to directly monitor the process of hydrophobic aggregation in bulk environment and further quantify hydrophobic interactions...”

- 4. Line 136: The authors state that the aggregation time scale is comparable to those of early stage protein folding. This however, strongly depend on the solute concentration. The authors should comment on how the concentrations they used were “equivalent” to hydrophobic residues on a polymer chain for fair comparison.**

Reply: We would like to thank the reviewer for this great comment. We have followed the reviewer’s suggestion to compare the HPS concentration in our experiment with that of residues on a protein chain. We estimated local concentrations of protein residues by computing the number of residues within a distance of the protein chain length (e.g. a protein domain containing 89-100 residues has a chain length of 28-30nm, see *Science*, 276, 1109 (1997)). The resulting local concentration of protein residues (1.3-1.8 mM) is comparable with our initial HPS concentration in the microfluidics tube: 2.0-6.0 mM. To include the above discussions, we have included the following sentences in the SI Sec. 6 of the revised manuscript, and revised Line136 to refer to this SI section:

“...The nucleation growth curve (Supplementary Fig. 7) suggests the microsecond time scale of HPS aggregation, which is comparable to the time scale of early stage protein folding. This comparison is made between similar concentration of HPS in our experiment and local concentration of residues on a protein chain. We estimated local concentrations of protein residues by computing number of residues within a distance of the protein chain length (e.g. a protein domain containing 89-100 residues has a chain length of 28-30nm). The resulting local concentration of protein residues (1.3-1.8 mM) is comparable with our initial HPS concentration in the microfluidics tube: 2.0-6.0 mM....”

- 5. Fig. 2b: It is unclear why the relative fluorescence of 3-6mM samples have non-zero y-intercepts. My understanding from the manuscript is that aggregation starts at time 0, and that no fluorescence should be observed before aggregation occurs, hence 0 fluorescence intensity at t=0. Are the authors simply o setting the traces to make the graph clearer? If so, it should be clearly stated and better illustrated.**

Reply: Yes, we have simply shifted the curves along y-axis to make clearer illustration. All the fluorescence curves should start out from 0 and t=0. In the revised manuscript, we have added the following sentence in the caption of Fig. 2 to explicitly state it. We apologize for any confusion it may have caused.

“...For clear illustrations, the relative fluorescence curves in part (b) corresponding to HPS concentrations of 3mM, 4mM, and 6mM are shifted along y-axis by 0.2, 0.4, and 0.6 respectively. Similarly, relative fluorescence curves in part (c) corresponding to DMSO mole fractions of 0.26 and 0.21 are shifted by 0.2 and 0.4 respectively. ...”

- 6. Fig. 2b: There is a weak dependency of the solute solvent surface tension from 14.5 to 14.2 on HPS concentration (6-2mM). Why does such dependency exist?**

Reply: We would like to thank the reviewer for his/her diligent reading of our manuscript. Since all these reported solute-solvent surface tension values are within the standard deviation $\pm 0.3 \text{ kcal/mol}$, we are uncertain if there exists a meaningful trend between solute-solvent surface tension and initial solute concentration.

7. Supp.Fig.4: One of the key assumptions is that the particle volume scales linearly to fluorescence, regardless of the size of the aggregate. The evidence provided is a linear plot of fluorescence intensity vs particle size from AFM measurement. In this experiment, the particle volume is on the order of 10^6 nm^3 , or $100 \times 100 \times 100 \text{ nm}$ (~1 million HPS molecules), these are huge aggregates and not surprising that total fluorescence is proportional to the volume. At this size scale, it is easy to extrapolate a fit close to 0. The size of aggregates formed in the flow are much smaller - between 2-10nm, or 5-100 molecules. Therefore, if the line does not intercept exactly at 0, the fluorescence-size linearity would fail for small aggregates. It is not immediately clear how the fluorescence of very small aggregates (2, 3, 4, 5 HPS molecules) are necessarily linearly correlated to the number of HPS in the aggregate.

Reply: We would like to thank the reviewer for this great comment. In our original manuscript, we provided three types of evidence to support the assumption that the total aggregate volume scales linearly with fluorescence intensity: the AFM experiment (Fig. S4), spectrophotometer experiments (Fig. S3), and quantum mechanics/molecular mechanics (QM/MM) calculations (Fig. S1, SI Ref. 34). As pointed out by the reviewer, both AFM and bulk spectrophotometer experiments were conducted under the condition of significantly larger HPS aggregate size compared to that in the microfluidic experiments. The QM/MM calculations show that when aggregates reach size 20 or larger, the fluorescence quantum efficiency (FQE) remains nearly invariant, indicating a linear relationship between aggregate volume and fluorescence intensity (see Fig. R6 (a) or Fig. S1 (c)). For very small aggregates (size below 20), we agree with the reviewer that it is not clear if this linear relationship would hold.

To address this issue, we computed the fraction of small aggregates (size below 20) in total aggregate volume when detectable fluorescence is present, and show that they have little impact: $< 3\%$ (see Fig. R6 b1-b7). This is consistent with the fact that in our experiments, we can only begin to detect fluorescence after ~ 6 microseconds, at which time the averaged aggregate size has already reached ~ 70 , a size that is significantly larger than 20 (under all HPS concentrations and DMSO fractions). Therefore, we believe that these small aggregates have negligible contributions to our fitting results. We would like to thank the reviewer again for this great comment.

Fig. R6. (a). The fluorescence quantum efficiencies (FQEs) of HPS amorphous aggregates with sizes of 10, 20, 30, 40 and 60 were compared. This figure is reproduced from *Nanoscale*, 8, 15173 (2016). (b)-(h). The fraction of the small aggregates ($n < 20$) in total aggregate volume as a function of time for seven experimental systems in our study. The black, blue, green and red curves in (b)-(e) represent the systems in the same solvent (with 0.16 mole fraction of DMSO) but with different initial HPS concentrations of 6 mM, 4 mM, 3 mM and 2 mM, respectively; while the black, magenta, cyan and yellow curves in (b), (f)-(h) represent the systems with the same HPS initial concentration (6 mM), but with different DMSO mole fractions of 0.16, 0.21, 0.26 and 0.32, respectively.

8. Equation 1: It wasn't clear how the authors took into account the DMSO concentration gradient along the jet in their nucleation growth model. Additionally, the DMSO concentration profile along the jet was calculated but not experimentally verified (Supp.Fig.9d). The authors should comment on the reliability of such calculation. One can imagine using a dye in the same experimental setup to validate such simulation.

Reply: We would like to thank the review for this great comment. We will first address the second part of this comment. In the microfluidics field, computational fluid dynamics (CFD) simulation has been shown to be a robust and reliable method to quantify mass transfer in fluidic system. During our CFD simulations, essential parameters including DMSO's diffusion coefficient, fluid viscosity, fluid density, and their variations in response to the changes of fluid composition have all been taken into account to obtain the DMSO concentration gradient along the jet. This numerical simulation approach has been well validated both qualitatively (e.g. *RSC Adv.*, 3, 17762, (2013) by us, and *ACS Nano*, 4, 2077, (2010) by Jahn *et al.*) and quantitatively

(e.g. *Anal. Chem.*, 87, 5589, (2015) by us) in various solvent mixtures. To include the above discussions, we have included the following sentences in the revised manuscript (SI Sec. 1):

“...CFD simulation has been shown to be a robust and reliable method to quantify mass transfer in fluidic system. During our CFD simulations, essential parameters including DMSO’s diffusion coefficient, fluid viscosity, fluid density, and their variations in response to the changes of fluid composition have all been taken into account to obtain the DMSO concentration gradient along the jet. This numerical simulation approach has been well validated both qualitatively (e.g. by us⁶ and Jahn et al.⁷) and quantitatively⁴ in various solvent mixtures...”

We are grateful for the reviewer’s first part of this comment. In our original CNT model, we actually treat the DMSO concentration as a constant. However, as pointed out by the reviewer, the DMSO concentration profiles display noticeable drift along the jet. To address this comment, we have modified our theory to incorporate the DMSO concentration variation along the microfluidic jet. To achieve this, we first fitted each DMSO concentration profile with a double-exponential function (see the updated SI Fig. 9d). We then applied these functions of DMSO concentrations to our nucleation growth model. In particular, the phase transfer free energy ($-h$), super-saturation ratio (S), and all other quantities depending on these two parameters become DMSO concentration dependent (see SI Sec. 11 for details). As shown in Fig. R7 (b-c), the modified model considering DMSO concentration gradient produced consistent results with our original model in both hydrophobic free energies ($\Delta\Delta G$) and cooperativity. These results suggest that taking into account the DMSO concentration gradient does not have significant impact on the fitting results. To further explain this, we show that $-h$ only has minor shift ($\sim 2\%$) when the DMSO concentration is allowed to vary, and the $-h$ value from our previous model lies well within the standard deviation of that from this modified model (see Fig. R7 (a)). As a result, $\Delta\Delta G$ predicted from our original model is also well within the standard deviation of the modified model when considering DMSO concentration gradient (see Fig. R7 (b)). For the modified model, we also noticed that the quality of fitting is noticeably reduced for certain systems (e.g. larger root mean square errors for system *e*, *g* and *f* as shown in Fig. R8) in comparison to the original model, which may be due to the increase of numerical complexity upon the introduction of a double-exponential function for DMSO concentrations.

In the revised manuscript, we have included the following sentences in the maintext and added a new section in SI (SI Sec. 11) to describe the above model that considers the DMSO concentration change. In addition, we have included Fig. R7 and R8 as SI Fig. 22 and SI Fig. 23, and also updated SI Fig. 9d.

Fig. R7. (a). Comparison of phase transfer free energy ($-h$) computed from the original model with constant DMSO concentration (in blue) and a modified model considering DMSO concentration change along the microfluidic tube (in red). The uncertainty of the blue bar was estimated from the standard deviations of $-h$ values along the microfluidic tube. (b) and (c) show $\Delta\Delta G$ and cooperativity, respectively for the original model (blue curves and bars) and the modified model (red curves and bars). The uncertainties of blue curves were computed based on the uncertainty of $-h$ reported in (a).

Fig. R8. (a) Root mean square (RMS) errors of theoretical fitting with respect to experiment in normalized fluorescence intensities for seven experimental systems in our study (b-h). The results of theoretical fitting from the original model with constant DMSO concentration (in grey) and a modified model considering DMSO concentration change along the microfluidic tube (in blue) are compared. (b-h) shows the fluorescence measured by the experiments (light dots) and predicted by the theory considering DMSO concentration change (solid curves). The black, blue, green and red curves in (b)-(e) represent the systems in the same solvent (with 0.16 mole fraction of DMSO) but with different initial HPS concentrations of 6 mM, 4 mM, 3 mM and 2 mM, respectively; while the black, magenta, cyan and yellow curves in (b), (f)-(h) represent the systems with the same HPS initial concentration (6 mM), but with different DMSO mole fractions of 0.16, 0.21, 0.26 and 0.32, respectively. To obtain converged numerical solutions, the first $5\mu\text{s}$ of experimental fluorescence data was not included in the fitting. Please refer to SI Sec. 11 for more details of the modified model considering DMSO concentration change along the microfluidic tube.

9. Lastly, it would be nice if the authors could show how the cooperativity of hydrophobic contribution is dependent on temperature and isolate the entropic vs enthalpic components.

Reply: We totally agree with the reviewer that it will be very interesting if one could elucidate the temperature dependence of hydrophobic cooperativity and further dissect the contributions from entropy and enthalpy. We feel that it is out of scope of the current manuscript. One of our major future directions is to set up proper temperature control apparatus to explore the temperature dependence of hydrophobic interactions. We have included the following sentence to discuss these future perspectives in the revised manuscript:

“...We expect that our experimental platform will have promising applications in studying hydrophobic interactions of a wide range of organic molecules with aggregation-induced emission⁵⁶, and in investigating the impact of important factors such as temperature on hydrophobic effect....”

Response to Reviewer #3's comments

We appreciate this reviewer's acknowledgement of the novel contributions made by our work to the understanding of hydrophobic interactions and assembly. This reviewer also thinks that our manuscript will find broad audience in both experimental and theoretical community. Following this reviewer's suggestion, we are planning a follow-up long paper to include more technique details. This reviewer also raised a number of helpful comments for us to further polish our manuscript particularly on the presentation of our work.

- 1. In the abstract and in the introduction the authors seem to imply that hydrophobic interactions are "fundamental" like ionic, dipolar, and dispersion forces. This is simply not true. The hydrophobic interaction is at its heart a thermodynamic force that arises after non-polar moieties are placed in an aqueous medium. Hydrophobic interactions do not exist outside of this context. Ionic, dipolar, and dispersion interactions, on the other hand, exist in all contexts in which there are molecules involved, arising from the arrangements of electrons about atomic nuclei. In this case then, I would consider ionic, dipolar, and dispersion interactions to be fundamental, while hydrophobic interactions. Rather hydrophobic interactions arise from the contortions water has to take about non-polar species in order to accommodate non-polar solutes.**

Reply: We agree with the reviewer that hydrophobic interactions only exist when non-polar solutes are introduced in aqueous solutions, which is still a thermodynamic force. This is distinct from fundamental inter-molecular forces introduced in General Chemistry textbook, which are all originated from electron arrangements surrounding atomic nuclei and exist in all contexts. To clarify this point, we have revised a few sentences in the maintext as follows:

In abstract: "Hydrophobic interaction is one of the important intermolecular forces....".

In Introduction: "... This is in contrast to fundamental intermolecular interactions that are often treated as pairwise additive such as ionic interactions, dipolar interactions, and dispersion forces...".

- 2. The authors imply in the introduction that only hydrophobic interactions are cooperative, while the others are only pairwise additive. This is approximately true, but there are significant efforts trying to incorporate polarizability and multi-body effects into simulations, which are clearly not pairwise additive. Moreover, pairwise additive interactions in protein folding can also give rise to cooperative effects. The most prominent example of this is the formation of protein helices by hydrogen-bonding down the peptide backbone. Helix melting curves point to a cooperative nature, which has been successfully captured by the Zimm-Bragg and Lifson-Roig helix-coil models which treat hydrogen-bonding as simply pairwise additive. It would be hard in my mind to try to disentangle the cooperativity the authors investigate from other sources of cooperativity in a protein folding experiment. This work does not effect that. I would also think that part of the cooperativity in proteins folding arises from the spatial locality of the aggregating units along the peptide backbone, not dispersed molecules coming together to form an aggregate. In this case the time scales for protein folding and the present experiments would not be comparable..**

Reply: We would like to thank the reviewer for this good comment. We agree with the reviewer that our expression that “the fundamental inter-molecular interactions are all pairwise additive” in the original manuscript is not entirely accurate. For example, when polarizability is considered, additional dipole may be induced depending on the surrounding electrostatic environments, and thus interactions between dipoles become not purely pairwise additive. Nevertheless, in common models, these interactions are often treated as pairwise additive. In the revised manuscript, we have modified the following sentences to avoid these inaccurate assertions:

In abstract: “... This is in contrast to other well-characterized fundamental intermolecular forces that are often treated as pairwise additive such as ionic interactions, dipolar interactions, and dispersion forces....”

In introduction: “... This is in contrast to fundamental intermolecular interactions that are often treated as pairwise additive such as ionic interactions, dipolar interactions, and dispersion forces....”

We also agree with the reviewer that protein folding is a complicated process that may involve different sources of cooperativity. As pointed out by the reviewer, Zimm-Bragg (*JCP*. 31, 526 (1959)) and Lifson-Roig models, which treat intra-chain hydrogen bonding as pairwise additive interactions, can be applied to successfully predict a cooperative helix melting process (e.g. *JACS*, 131, 2306, (2009)). Furthermore, cooperativity may also arise when many amino acid side-chains are brought together in spatial proximity upon protein folding as speculated by the reviewer. Nevertheless, we believe that our reported cooperativity has implications on part of cooperativity in globular protein folding that arises from the formation of hydrophobic cores, as both processes are underlined by water-mediated hydrophobic interactions. To include the above discussions, we have included the following sentences in the maintext of the revised manuscript:

“...We acknowledge that these two processes are different in many aspects. For instance, flexible protein chains contain numerous conformations, while HPS molecules are relatively rigid. In addition, multiple sources may contribute to cooperativity in protein folding, such as the cooperative helix melting process due to hydrogen bonding⁵⁵...”

3. There is no indication in the introduction what the authors are going to do to place how their contribution would fit into the wider knowledge base on hydrophobic interactions. Rather than authors jump into the description of what their contribution is starting at the beginning of the results and discussion section. While I am not an expert on microfluidic mixing, surely this has been done before. It seems like it would be wise in the introduction discuss how this technique applies to the problem at hand. The way this reads now is just jarring.

Reply: We agree with the reviewer and have followed his/her suggestion to include the following sentences to review the current applications of the microfluidics mixing technique.

“... Microfluidic mixing techniques have been utilized to investigate many important chemical and biological processes, including protein and RNA folding, enzyme activities, and vesicle formations... where the molecular aggregation occurs in a sample stream that was

hydrodynamically sheathed to tens of nanometers in width within a few microseconds upon rapid solvent exchange. ...”

We also included the following sentence in the maintext to better describe where our contributions the context of existing studies aiming to quantify hydrophobic interactions in the bulk environment:

“...Although earlier experimental studies have attempted to estimate water-mediated hydrophobic interactions, it remains challenging to directly monitor the process of hydrophobic aggregation in bulk environment and further quantify hydrophobic interactions...”

4. The authors do point to previous theoretical work on large scale hydrophobic interactions, but they do not take steps to try to connect their work to those efforts. 2 In the case of aggregates this suggests that a kink would arise if the free energy for forming a cluster divided by the aggregation number raised to the 2/3's power ($N^{2/3}$) is proportional to the aggregate surface area) was plotted against $N^{1/3}$ (proportional to the aggregate radius). Cooperativity in the LCW theory then arises from the differences in scaling of the free energy on one side of the kink versus the other. It would be instructive if the authors attempted this type of calculation to touch base with existing theoretical/scaling ideas associated with hydrophobic interactions.

Reply: We would like to thank the reviewer for this insightful comment! It is an excellent idea to make connections between cooperativity measured from our experiment with the LCW theory by comparing the solvation free energy per surface area ($\Delta G/\text{\AA}^2$) *before* and *after* the crossover (or the “kink”) in the aggregate radius (a similar plot as Fig. 2 in Chandler, *Nature*, 437, 640, (2005)). The magnitude of the cooperativity should then be correlated with the difference in $\Delta G/\text{\AA}^2$. As shown in Fig. R9, we plotted the $\Delta G/\text{\AA}^2$ for the HPS aggregation in four different DMSO/water mixture solvents. Indeed, there exists a clear kink before and after the crossover length scale at around 1 nm as suggested by the LCW theory. More interestingly, the difference in $\Delta G/\text{\AA}^2$ before and after the kink clearly increases with the increase of water fraction in the solvent mixture, indicating stronger cooperative part of the formation energy in solvent mixtures containing more water.

Fig. R9. The solvation free energy (ΔG_A) per solute surface area as a function of the solute radius. The black, gray, magenta, blue and green dots represent the systems with DMSO mole fraction of 0, 0.16, 0.21, 0.26 and 0.32, respectively. The first point at $R = 5.9 \text{ \AA}$ corresponds to the solvation free energy an individual HPS molecule (h) divided by its solvent accessible surface area (750 \AA^2). The other points correspond to HPS aggregates with size three and above, and their hydration free energies and surface area were obtained from $\theta n^{2/3}$ and $4\pi(R_n + R_W)^2$ respectively. Please refer to SI Sec. 10 for calculation details. The dashed lines are the extrapolation of all the points after the crossover of the length scale (aggregates of size 3 and above with $R > 10 \text{ \AA}$). These lines are all above the first point before the crossover ($R = 5.9 \text{ \AA}$), indicating a kink before and after the crossover of the length scale as predicted by the LCW theory.

To include the above discussions, we have added the following sentences in the maintext of the revised manuscript. In addition, we added a new section in SI. (Sec. 10: Connecting with the LCW theory), and inserted Fig. R9 as SI Fig. 21 in the revised manuscript. Once again, we are grateful for this great suggestion from the reviewer.

“...Interestingly, Chandler and co-workers predicted that there should exist a crossover in the length-scale for the hydrophobic effect at around 1 nm in solute radius^{1,50}. Our experiment demonstrates the existence of this crossover by showing a kink during the transition from volume-based hydration free energy for monomer ($< 1 \text{ nm}$ in radius) to area-based hydration free energy of aggregates ($> 1 \text{ nm}$ in radius, see Supplementary Sec. 10 and Supplementary Fig. 21 for details)...”

5. I do wonder if the cooperativity described here is even surprising. As the aggregates get bigger and bigger they are effectively form a pure HPS phase. Phase separation is certainly a cooperatively process because there is no new phase unless a lot of things came together to make that phase. When I look at the data in Figure 1c then I find myself wondering if the authors have simply just determined the free energy of transferring a single HPS into its neat liquid (? – or solid). It this known from independent measurements? I know Walker and Li demonstrated the correlation between their pulling experiments with solubility measurements – has any such correlations been attempted here?

Indeed the cooperativity of micelle formation has long been associated with the free energy of transferring surfactant hydrocarbon tails into a neat liquid phase – the so-called phase separation model of micellization. I can’t tell if the authors have adding anything more beyond this model for assembly, which is quite old and accurate.

Reply: We would like to thank the reviewer for this good comment. The transfer free energy of a single HPS from water into the HPS phase as pointed out by the reviewer is actually the large n limit of our reported hydrophobic free energies: $\lim_{n \rightarrow \infty} \Delta \Delta G$. In our model, this phase transfer free energy ($-h$) is obtained from independent solubility measurements (see SI Eq. 15 and Fig. S10), and has been applied to derive $\Delta \Delta G$ (see SI Eq. 14). As shown in Fig. 1c, hydrophobic free energies ($\Delta \Delta G$) vary with the size of the aggregate, e.g. When $n = 100$, $\Delta \Delta G = -11.1 \text{ kcal/mol}$, which is still substantially different from the macroscopic phase transfer free energy: $\lim_{n \rightarrow \infty} \Delta \Delta G = -h = -13.6 \text{ kcal/mol}$. Therefore, our measurements can provide values of hydrophobic free energy over a wide range of the aggregate size, which cannot be accessed by the macroscopic measurement of the phase transfer free energy. Furthermore,

our experiment also allows us to estimate the pair potential of mean force, which is a critical parameter to obtain the cooperativity of hydrophobic interactions. Therefore, we think that the novelty of our experiment lies in its capability to provide hydrophobic free energies for aggregates with finite sizes (not restricted to the large size limit), as well as cooperativity associated with their formation.

- 6. The authors seem to want to compare their aggregation free energies against the energy of water forming a hydrogen-bond. I realize this comes about because of the cartoon picture put forward in previous work that the formation of large surfaces is accompanied by the loss of hydrogen bonds. Water would already not be able to form a complete hydrogen-bonding network about a single HPS model with its molecular detail, nooks and crannies. Aggregation in this case is not likely to change the extent of water hydrogen bonding as the aggregates get bigger and bigger. Rather this is likely a surface tension driven effect. The crossover from volume to surface area based interactions in the LCW theory is not simply a result of hydrogen bonding. Indeed LCW theory has no hydrogen bonding in it. Rather the crossover occurs as a result of density fluctuations changing from approximately Gaussian in nature for small scale solutes, to distinctly non-Gaussian for larger aggregates as a result of the thermodynamic proximity of the solvent (water in this case) to a liquid-vapor phase transition. The solvent then does not have to have hydrogen-bonding, but simply be close to a phase transition. Water is unique amongst solvents, however, do to its larger surface tension (which may be argued arises from hydrogen bonding). Nevertheless, the cooperativity the authors report likely is not a result of hydrogen-bonding but the closeness to water's vaporization transition.**

Reply: We agree with the reviewer that a direct explanation for the crossover of the hydrophobic effect is that waters around large non-polar solutes undergo significant non-Gaussian density fluctuations, and thus render them in proximity to a liquid-vapor phase transition. This phenomenon has been suggested to be related to the persistence of the hydrogen-bonding network. For example, Lum *et al.* suggests that a drying transition will occur around large non-polar solutes due to an energetic effect, given that the persistence of the hydrogen bond network becomes geometrically impossible (*JPCB* 103, 4570 (1990)). To include the above discussions, we have included the following sentences in the maintext and SI Sec. 10:

“... They suggested that this crossover is related to the persistence of the hydrogen-bonding network. For example, a drying transition may occur around large non-polar solutes due to an energetic effect, given that the persistence of the hydrogen bond network becomes geometrically impossible³⁸. A more direct explanation for the crossover of the hydrophobic effect is that waters around large non-polar solutes undergo significant non-Gaussian density fluctuations, and thus render them in proximity to a liquid-vapor phase transition....”

- 7. Beginning on line 120 the authors state “As hydrophobic interaction is a major driving force for protein folding, proteins will be likely unable to fold into their three-dimensional structures in the absence of hydrophobic cooperativity.” This statement cannot be concluded from the HPS aggregation measurements performed herein. Indeed folded protein conformations are stabilized by energies on the order of 10 kcal/mol. The authors find aggregate formation free energies much larger than this for aggregates as small as 10 HPS's (see line 97). The free energies for stabilizing proteins**

thus are considerable smaller than what they report in the cooperative regime for larger aggregates. Cooperative boasts to the folding free energy do not seem to be necessary for protein folding based on the authors own results.

Reply: We agree with the reviewer that the abovementioned sentence is not entirely accurate. We decided to remove it in the revised manuscript. Indeed, the aggregation of dispersed HPS molecules is substantially different from folding of flexible protein chains. For example, these two processes are associated with different entropic contributions because flexible protein chains contain numerous conformations, while our HPS molecules are small and relatively rigid. In addition, protein folding is a complicated process that may involve different sources of cooperativity.

8. Moreover, hydrophobic interactions are not thought to stabilize the three-dimensional structure of proteins. Rather these interactions are thought to help collapse the chain into a globular shape. Three dimensional secondary structure formation however certainly need hydrogen bonding and side chain packing to arrive at its final shape.

Reply: We agree with the reviewer's comment. We have revised the statement as follows:

"...Hydrophobic interaction plays a crucial role in facilitating the collapse of protein chains into a globular shape^{25,52-54} ..."

9. Line 143. The authors write "nature designs hydrophobic interaction" is not true. Nature doesn't design the hydrophobic interaction. It might make use of it and take advantage of it to its own ends, but it doesn't design it.

Reply: We have removed the word "design" here and rewritten the sentence as follows:

"... Most importantly, we show that hydrophobic interaction, an interaction induced by collective behaviors of many water molecules, is strongly cooperative (as large as 40%) and thus substantially enhance its strength during the aggregation of dispersed hydrophobic molecules in solution..."

10. Line 145. The authors write "Without cooperativity, proteins may not be able to fold to specific structures, and as a consequence all life formed on Earth may perish." My jaw just dropped when I read this concluding sentence. This sentence reads like we need to do something to make sure hydrophobic interactions remain cooperative or we are all going to die. This is a bit of a grandiose misstatement. We are not in danger of losing cooperativity. Without the ability for proteins to fold, the more appropriate statement would be that life as we know it would not have appeared. Since we are here, however, it looks like the hydrophobic effect is working just fine and we are not imperiled. This is a gross oversell.

Reply: We agree with the reviewer and have removed this sentence.

11. The whole concluding paragraph on page 4 has almost nothing to do with the rest of the paper. We already knew that hydrophobic aggregation was faster than the final protein folding, so if the only conclusion of this paper is something we already knew. There is no summary of the salient conclusions that can be drawn from this work and the

extrapolations are grandiose and do not directly follow from the rest of the paper. This whole paragraph should be written in a thoughtful manner.

Reply: We have largely revised the concluding paragraph as follows:

“Organic molecules, such as proteins and lipid, bury their hydrophobic components to form stable cores. Hydrophobic interaction plays a crucial role in facilitating the collapse of protein chains into a globular shape^{25,52-54}. The faster kinetics of hydrophobic aggregations (at microsecond), in contrast to protein folding (at millisecond or longer), suggest that the formation of protein cores by the aggregation of hydrophobic side-chains occurs at the early stage in the process of globular protein folding. Most importantly, we show that hydrophobic interaction, which is an interaction induced by collective behaviors of many water molecules, is strongly cooperative, and thus substantially enhance its strength during the aggregation of dispersed hydrophobic molecules in solution. We anticipate that our findings have profound implications in protein folding, as the protein core formation involves the collapse of hydrophobic side-chains. We acknowledge that these two processes are different in many aspects. For instance, flexible protein chains contain numerous conformations, while HPS molecules are relatively rigid. In addition, multiple sources may contribute to cooperativity in protein folding, such as the cooperative helix melting process due to hydrogen bonding⁵⁵. In spite of these differences, our findings highlight the important role of hydrophobic cooperativity (as large as 40%) in the initial collapse of protein chain to form into a globular shape. We expect that our experimental platform will have promising applications in studying hydrophobic interactions of a wide range of organic molecules with aggregation-induced emission⁵⁶, and in investigating the impact of important factors such as temperature on hydrophobic effect.”

12. Line 12. Insert a “The” before “hydrophobic” in the first sentence of the abstract. Also the hydrophobic interaction is not fundamental (see above).

Reply: We have revised the manuscript accordingly.

13. Line 34. Awkward wording “... of hydrophobic effect ...”. Sentence needs to be rewritten.

Reply: We have rewritten “hydrophobic effect” into “hydrophobicity”.

14. Line 48. “... and insofar focused ...”. Insofar means “to the extent that” so this statement makes no sense.

Reply: We have rewritten “insofar” into “so far has”.

15. Line 120. “As hydrophobic interaction is a major ...”. Awkward sentence beginning. Not grammatical.

Reply: We have removed this sentence.

**16. Line 123. “... at tenths to hundredths of nanoseconds.” I think they mean tens to hundreds. Tenths and hundredths of nanoseconds are 1/10 and 1/100 nanoseconds. -
Line 129. “Slows” not “slow”**

Reply: We have revised the manuscript as suggested.

17. Line 138. I know biochemists might call lipids macromolecules, but they are not.

Reply: “Macromolecules” now changed to “organic molecules”

18. Line 143. “the hydrophobic interaction” not “hydrophobic interaction”.

Reply: We have revised the manuscript as suggested.

REVIEWERS' COMMENTS:

Reviewer #1 (Remarks to the Author):

I am generally satisfied with the substantial effort that the authors have expended in addressing my (reviewer 1) comments , as well as those of the other two reviewers. My only remaining concern pertains to my comment (2), as the authors reply to that comment has missed the point that the paper by Harris and Pettitt (2016), and references therein, have brought into question the long held assumption that hydrophobic interactions play a critical role in protein folding. In other words, the validity of that assumption is now an open question. I suggest that the authors make this clear clear when they cite the above reference.

Although this manuscript is not the last word in addressing numerous open questions regarding the nature of hydrophobic interactions, it is an important contribution that I believe is likely to draw significant attention and inspire subsequent studies.

Reviewer #2 (Remarks to the Author):

The authors have done an exceptionally thorough job addressing my original concerns with the manuscript. In my view, the manuscript provides a nice system to study specific hydrophobic interactions involving fluorophores, and can establish the experimental basis of downstream theoretical studies. The authors have included significant additional data and edits to the text to strengthen this manuscript. Therefore, I recommend the publication of this manuscript.

Response to Reviewer #1's remaining comment

1. My only remaining concern pertains to my comment (2), as the authors reply to that comment has missed the point that the paper by Harris and Pettitt (2016), and references therein, have brought into question the long held assumption that hydrophobic interactions play a critical role in protein folding. In other words, the validity of that assumption is now an open question. I suggest that the authors make this clear when they cite the above reference.

Reply: We have followed the reviewer's suggestion to include the following sentence in the maintext while citing *Harris and Pettitt (2016)* (J. Phys.: Condens. Matter, 28, 083003 (2016)) to make it clear that the critical role of hydrophobic interactions to protein folding has recently been brought into question:

"...Furthermore, even the extent of contributions by hydrophobic interactions to protein folding remains elusiv⁴⁹..."